# Effects of $NO_2$ and $C_3H_6$ on the heterogeneous oxidation of $SO_2$ on $TiO_2$ in the presence or absence of UV-Vis irradiation

Biwu Chu [1, 2, 3], Yali Wang [1, 3], Weiwei Yang [1, 3, 4], Jinzhu Ma [1, 2, 3], Qingxin Ma [1, 2, 3, *], Peng Zhang [1], Yongchun Liu [1, 5], Hong He [1, 2, 3]

[1] State Key Joint Laboratory of Environment Simulation and Pollution Control, Research Center for Eco-Environmental Sciences, Chinese Academy of Sciences, Beijing 100085, China
[2] Center for Excellence in Regional Atmospheric Environment, Institute of Urban Environment, Chinese Academy of Sciences, Xiamen 361021, China
[3] University of Chinese Academy of Sciences, Beijing 100049, China
[4] Key Laboratory of Pesticide and Chemical Biology of Ministry of Education, Institute of Environmental and Applied Chemistry, College of Chemistry, Central China Normal University, Wuhan 430079, China
[5] Aerosol and Haze Laboratory, Beijing Advanced Innovation Center for Soft Matter Science and Engineering, Beijing University of Chemical Technology, Beijing 100029, China

*Correspondence to*: Qingxin Ma (qxma@rcees.ac.cn)

**Abstract.** The heterogeneous reactions of $SO_2$ in the presence of $NO_2$ and $C_3H_6$ on $TiO_2$ were investigated with the aid of *in situ* Diffuse Reflectance Infrared Fourier Transform Spectroscopy (DRIFTS) under dark conditions or with UV-Vis irradiation. Sulfate formation with or without the coexistence of $NO_2$ and/or $C_3H_6$ was analyzed with IC. Under dark conditions, $SO_2$ reacting alone resulted in sulfite formation on $TiO_2$, while the presence of ppb levels of $NO_2$ promoted the oxidation of $SO_2$ to 20 sulfate. The presence of $C_3H_6$ had little effect on sulfate formation in the heterogeneous reaction of $SO_2$ but suppressed sulfate formation in the heterogeneous reaction of $SO_2$ and $NO_2$. UV-Vis irradiation could significantly enhance the heterogeneous oxidation of $SO_2$ on $TiO_2$, leading to copious generation of sulfate, while the coexistence of $NO_2$ and/or $C_3H_6$ significantly suppressed sulfate formation in experiments with UV-Vis lights. Step-by-step exposure experiments indicated that $C_3H_6$ mainly competes for reactive oxygen species (ROS), while $NO_2$ competes with $SO_2$ for both surface active sites and ROS. 25  Meanwhile, the coexistence of $NO_2$ with $C_3H_6$ further resulted in less sulfate formation compared to introducing either one of them separately to the $SO_2$-$TiO_2$ reaction system. The results of this study highlighted the complex heterogeneous reaction processes that take place due to the ubiquitous interactions between organic and inorganic species, and the need to consider the influence of coexisting VOCs and other inorganic gases in the heterogeneous oxidation kinetics of $SO_2$.

## 1 Introduction

Atmospheric aerosol pollution has attracted widespread attention in recent years because of its adverse effects on human health, visibility and climate (Thalman et al., 2017;Davidson et al., 2005;Pöschl, 2005). In many developing countries, such as China and India, high concentrations of $SO_2$, $NO_x$, and volatile organic compounds (VOCs) coexist in the atmosphere (Zou et al.,

2015;Liu et al., 2013;Yang et al., 2009) and result in "complex atmospheric pollution" (Yang et al., 2011) and heavy haze events. Sulfate was found to play important roles in the occurrence of these haze events (Zhang et al., 2011;Liu et al., 2017b) due to both its high mass concentration in fine particles ($PM_{2.5}$) and its strong hygroscopicity. Rapid formation of sulfate was frequently observed in haze episodes in China, in which heterogeneous reactions played important roles (He et al., 2014;Zhang et al., 2006;Ma et al., 2018). However, the mechanism of the heterogeneous reaction process as well as its contribution to sulfate formation in "complex atmospheric pollution" remain uncertain (Yang et al., 2018;Ma et al., 2018;Wang et al., 2018;Yu and Jang, 2018). These uncertainties are considered to be the main reason for the inaccuracy of sulfate simulation in air quality models (Wang et al., 2014b;Zheng et al., 2015;Yu and Jang, 2018).

About 1000 to 3000 Tg of mineral aerosols are emitted into the atmosphere every year (Dentener et al., 1996;Shen et al., 2013;Jaoui et al., 2008) and provide abundant surface area for the heterogeneous oxidation of $SO_2$. The heterogeneous uptake of $SO_2$ can form bisulfite ($HSO_3^-$) or sulfite ($SO_3^{2-}$) on $\gamma$-$Al_2O_3$ and sulfate ($SO_4^{2-}$) on MgO (Goodman et al., 2001a). Similarly, $SO_2$ can be converted into sulfite, bisulfite or sulfate on mineral dust such as metal oxides (Zhang et al., 2006), calcite, and China loess (Usher et al., 2002). The heterogeneous reaction of $SO_2$ on mineral dust can be promoted by gaseous oxidants. For example, $SO_2$ could be oxidized into sulfate by $O_3$ on the surface of $CaCO_3$ particles (Li et al., 2006;Zhang et al., 2018). Similar results were obtained when introducing $H_2O_2$ into the heterogeneous oxidation system (Capaldo et al., 1999;Jayne et al., 1990). $NO_2$ can also promote the heterogeneous oxidation of $SO_2$. In our previous studies, it was found that $SO_2$ was oxidized to sulfate on $\gamma$-$Al_2O_3$ in the presence of $NO_2$ and $O_2$, while it was only converted to sulfite in the absence of them (Ma et al., 2008). Therefore, $NO_2$ was proposed to act as a catalyst in the oxidation of $SO_2$ by $O_2$, in which the intermediates observed in the spectra, i.e. nitrogen tetroxide ($N_2O_4$), might play an important role (Ma et al., 2008). This synergistic effect between $SO_2$ and $NO_2$ was further observed on many other mineral oxides such as CaO, $\alpha$-$Fe_2O_3$, ZnO, MgO, $\alpha$-$Al_2O_3$, and $TiO_2$ (Liu et al., 2012;Ma et al., 2017;Zhao et al., 2018;Yu et al., 2018). These effects were confirmed in smog chamber studies and field observations of heavy haze in China, and were proposed to be an important reason for the rapid growth of sulfate in haze events (He et al., 2014;Ma et al., 2018;Wang et al., 2014a;Chu et al., 2016). Heterogeneous oxidation of $SO_2$ may also be affected by the coexistence of organic compounds. Pre-adsorption of acetaldehyde ($CH_3CHO$) was found to suppress the heterogeneous reaction of large amounts of $SO_2$ on the surface of $\alpha$-$Fe_2O_3$ (Zhao et al., 2015), while HCHO was proposed to react with $SO_3^{2-}$ and generate hydroxymethanesulfonate (HMS) in the northern China winter haze period (Moch et al., 2018;Song et al., 2019). Wu et al. (2013) found that the synergistic effects between formic acid (HCOOH) and $SO_2$ in the heterogeneous reaction on hematite provide a new source of sulfate.

UV illumination can affect both the properties of particles and heterogeneous reactions on them (Nanayakkara et al., 2012;Cwiertny et al., 2008;George et al., 2015). The photooxidation of $SO_2$ in the presence of mineral dust may represent an important pathway for generating sulfate aerosols (Park et al., 2017;Yu and Jang, 2018). $TiO_2$, an n-type semiconductor material, has been widely used for studying heterogeneous photochemical reactions (Chen et al., 2012). $TiO_2$ can be excited by UV light ($\lambda < 387$ nm), resulting in electrons and holes, which could react with $O_2$ and $H_2O$ and produce $\cdot O_2^-$ and $\cdot OH$, respectively. These reactive oxygen species (ROS), primarily $\cdot O_2^-$ and $\cdot OH$, can participate in the heterogeneous oxidation of

SO$_2$ on TiO$_2$ (Chen et al., 2012). Shang et al. (2010a) studied the heterogeneous reaction of SO$_2$ on TiO$_2$ particles using *in situ* Diffuse Reflectance Infrared Fourier Transform Spectroscopy (DRIFTS), and observed that SO$_2$ was oxidized to sulfate on TiO$_2$ with UV illumination while remaining as sulfite under dark conditions. Our recent study showed that O$_2$ and H$_2$O have contrary roles in the photooxidation of SO$_2$ on TiO$_2$, where surface water exhibits a competition effect in the reaction of

SO$_2$ due to the occupation of surface OH (Ma et al., 2019). Besides H$_2$O, the co-existence of organics may also suppress the formation of sulfate due to competition with SO$_2$ for reactive oxygen species. For example, Du et al. (Du et al., 2000) studied the photocatalytic reaction of SO$_2$ in the presence of heptane (C$_7$H$_{16}$) and found that the formation of sulfate was suppressed.

Despite these studies involving the heterogeneous oxidation of SO$_2$ under various conditions, the effects of co-existing pollutants on the heterogeneous oxidation of SO$_2$ under both dark and illuminated conditions need further investigation.

Meanwhile, the interactions between organic and inorganic species in these heterogeneous processes at low concentrations are not fully understood. In this study, we focus on the effects of co-existing NO$_2$ and propene (C$_3$H$_6$) on the heterogeneous oxidation of SO$_2$ on TiO$_2$ under both dark and illuminated conditions with *in situ* DRIFTS. In order to better study the effects of NO$_2$ and C$_3$H$_6$ on the heterogeneous oxidation in a relatively complex oxidation system (with coexistence of multiple gases, in both dark and illuminated conditions), we chose TiO$_2$ due to the fact that it is a semiconductor material and a well-known

photocatalyst. TiO$_2$ has been widely reported to be present in airborne particulate matter (PM) (Chen et al., 2012). Although TiO$_2$ represents only a relatively small portion of the mass of PM and is less abundant than CaO, Fe$_2$O$_3$ or MgO, the TiO$_2$ particles are expected to provide important surfaces for heterogeneous photocatalysis of atmospheric gases due to their high photocatalytic activity, especially with the growing application of TiO$_2$ in human activities (Chen et al., 2012). Propene is selected as a representative VOC since it is the most abundant alkene compound in the atmosphere, and coexists with NO$_x$ in

vehicle exhaust emission (Wang et al., 2016a). Propene is widely used as an accelerator in photochemical reactions in some smog chamber studies (Jang and Kamens, 2001;Song et al., 2007). The relatively simple oxidation products and well understood oxidation mechanism of propene are also helpful in explaining our experimental results. Propene is selected also due to the high vapor pressure of its oxidation products, which normally do not generate condensed organic aerosol (Odum et al., 1996). However, we must point out that the heterogeneous reactivity depends greatly on the properties of the mineral

oxides, such as acid−base nature or redox properties (Tang et al., 2016;Yang et al., 2016;Yang et al., 2019), while different VOCs may also have quite different heterogeneous and photochemical reactivity. Investigating these processes on different mineral dust and authentic dust particles with different types of VOCs is needed in future studies. Rather than UV lights, a xenon light is used in this study to better simulate the solar ultraviolet radiation on the earth's surface. Generally, our study could be helpful for gaining a better understanding of the heterogeneous formation of sulfate under complex air pollution

conditions, in which abundant SO$_2$, NO$_x$, VOCs, and mineral dust coexist in the atmosphere.

## 2 Experimental section

### 2.1 Materials

TiO$_2$ (Degussa P$_{25}$) used in this study was a typical commercially available material, which contains 75% anatase and 25% rutile. It has been widely used in laboratory studies due to its good photocatalytic properties. The surface area of the material in this study was 50.50 m$^2$ g$^{-1}$, measured by an ASAP2010 BET apparatus with multipoint Brunauer-Emmett-Teller (BET) analysis. The average particle diameter was about 20 nm, determined by transmission electron microscopy (H-7500, Hitachi Inc.). For gases, N$_2$ (99.999% purity, Beijing Huayuan) and O$_2$ (99.999% purity, Beijing Huayuan) were introduced as synthetic air (80 % N$_2$ and 20 % O$_2$) in this study, while SO$_2$ (5.9 ppm in N$_2$, Beijing Huayuan), NO$_2$ (3.9 ppm in N$_2$, Beijing Huayuan) and C$_3$H$_6$ (5.9 ppm in N$_2$, Beijing Huayuan) were used as reactant gases.

### 2.2 Experimental methods

#### 2.2.1 *In situ* DRIFTS

*In situ* DRIFTS spectra were recorded on a Nicolet Nexus 670 FTIR equipped with a mercury cadmium telluride (MCT) detector, scanning from 4000 to 650 cm$^{-1}$ at a resolution of 4 cm$^{-1}$ for 100 scans. Before each experiment, the oxide sample was finely ground and placed into a ceramic crucible in the *in situ* chamber. Then the sample was pretreated at 503 K and atmospheric pressure for 120 min to remove adsorbed species in 100 mL min$^{-1}$ synthetic air. All the spectra are presented in the Kubelka-Munk (K-M) scale to improve the linearity of the dependence of signal intensity upon concentration (Armaroli et al., 2004). The UV-Vis irradiation was acquired with 500 W xenon light (CHF-XM35, Beijing Chuangtuo) and was introduced into the DRIFTS reaction cell via a UV optical fiber. The intensity of UV-Vis irradiation was measured as 478 µW cm$^{-2}$ by a UV Meter (Photoelectric Instrument Factory of Beijing Normal University). The wavelengths of the UV-Vis irradiation were measured to be in the range of 300-800 nm by a fiber optic spectrometer (BLUE-Wave-UVNb, Stellar Net Inc., USA), as shown in Fig. S1 in the Supplemental Information. The spectrum of the UV-Vis irradiation is comparable to the spectrum of solar irradiation on the earth surface, and therefore we think the UV-Vis irradiation used in this study may represent the conditions in the real atmosphere.

To investigate heterogeneous sulfate formation in complex atmospheric pollution, *in situ* DRIFTS was used to analyze the products on particle surfaces in the reactions under different conditions. Two series of *in situ* DRIFTS experiments were carried out in this study. For the heterogeneous reaction of SO$_2$ under different gas conditions, the TiO$_2$ sample was initially flushed with the synthetic air at a total flow rate of 100 mL min$^{-1}$ for 2 h. The temperature was 303 K and the relative humidity was less than 1% in all experiments. Then the background spectra were recorded when they showed little change with time. After that, gas reactants, such as 200 ppb SO$_2$, 200 ppb NO$_2$ and 200 ppb C$_3$H$_6$, were introduced to the gas flow and then passed through the reaction chamber for 12 h. These experiments were carried out under both dark and with UV-Vis irradiation conditions. The other series of experiments were step-by-step exposure experiments for further investigation of the effects of NO$_2$ and C$_3$H$_6$ on the heterogeneous oxidation of SO$_2$ with UV-Vis irradiation. The concentrations of reactants in the step-by-

step exposure experiments were changed from 200 ppb to 200 ppm to strengthen the signals of the products. These step-by-step exposure experiments all included three steps, namely, first exposing the particles to $NO_2$, $C_3H_6$, or both for 2 h, then flushing with air for 1 h, and finally exposing them to $SO_2$ for 2 h.

### 2.2.2 IC

Sulfate products on the powders after the *in situ* DRIFTS study were also measured quantitatively using ion chromatography (IC). The powders were firstly weighed, and placed in 8 ml transparent glass jars. After adding 5 ml ultrapure water (specific resistance ≥ 18.2 MΩ cm$^{-1}$) containing about 1% formaldehyde (50 μL) to inhibit the oxidation of sulfite to sulfate, the samples were then extracted by sonication at 303K for 120 minutes. After a standing time of 120 minutes, the obtained supernatant was passed through a 0.22 μm PTFE membrane filter and then was analyzed using a Wayee IC-6200 ion chromatograph equipped with a Thermo AS14 analytical column. An eluent of 3.5 mM $Na_2CO_3$ was used at a flow rate of 0.8 mL min$^{-1}$.

### 3 Results

### 3.1 Heterogeneous reaction of $SO_2$ under different conditions

### 3.1.1 Heterogeneous reaction of $SO_2$ on $TiO_2$

DRIFTS spectra for heterogeneous reaction of 200 ppb $SO_2$ on $TiO_2$ under dark conditions or with UV-Vis irradiation are shown in Fig. 1, while the vibrational frequencies of chemisorbed species formed on the surface of $TiO_2$ are listed in Table 1. In the dark experiment, the reaction products on the surface of $TiO_2$ were mainly sulfite. As shown in Fig. 1(a), the positive bands observed at 1098, 1078, and 1052 cm$^{-1}$ can be assigned to monodentate sulfite (Hug, 1997;Peak et al., 1999). Negative peaks at 3691 and 3630 cm$^{-1}$ were attributed to hydroxyl on $TiO_2$ (Primet et al., 1971;Tsyganenko and Filimonov, 1973;Ferretto and Glisenti, 2003). These negative peaks were observed in all the reaction systems in this study, as shown in Fig. 1, which is consistent with previous studies (Nanayakkara et al., 2012;Ma et al., 2019). The loss of surface hydroxyl groups from the surface upon adsorption of $SO_2$ implies that surface OH groups were involved in the reaction of $SO_2$ on $TiO_2$ under both dark and UV-Vis irradiation conditions.

With UV-Vis light illumination, $SO_2$ was oxidized on $TiO_2$ and resulted in abundant sulfate species, as shown in Fig. 1(b). The main bands in the 1400-1100 cm$^{-1}$ region became more apparent with increasing exposure time. The spectra in this region were assigned to sulfate in different coordination modes, including aggregation at 1344 cm$^{-1}$, bidentate at 1290 cm$^{-1}$ and bridging sulfate at 1177 and 1141 cm$^{-1}$ (Hug, 1997;Peak et al., 1999;Fu et al., 2007). With UV-Vis illumination, $TiO_2$ can be excited by UV light ($\lambda$< 387 nm), then the photogenerated electrons and holes can react with $H_2O$ and $O_2$ to produce additional ROS (primarily $\cdot O_2^-$ and $\cdot OH$), and oxidize more $SO_2$ to sulfate on $TiO_2$ than that produced under dark conditions (Shang et al., 2010a;Chen et al., 2012). The sharp band at 1626 cm$^{-1}$ and the broad bands with maxima at 3316 and 3190 cm$^{-1}$ in Fig. 1(b) can be assigned to the bending vibration and stretching modes of molecularly adsorbed water. Surface water can

be formed in the heterogeneous reaction of $SO_2$ (Nanayakkara et al., 2012;Zhang et al., 2006), or via enhanced adsorption of water due to the increased hygroscopicity induced by sulfate (Ma et al., 2019). Although the RH was controlled at less than 1% in our experiments, water cannot be entirely removed in the introduced gas flows. In Fig.1, there is a positive correlation between the signal intensities of the adsorbed water and sulfite/sulfate among different experimental systems.

### 3.1.2 Heterogeneous reaction of $SO_2$ and $NO_2$ on $TiO_2$

As reported in previous studies, the presence of $NO_2$ can promote the heterogeneous oxidation of $SO_2$ (Ma et al., 2008;Liu et al., 2012;Ma et al., 2017), which was also investigated in this study under both dark and illuminated conditions. The spectra regarding the reaction of 200 ppb $SO_2$ and 200 ppb $NO_2$ on $TiO_2$ under dark conditions are shown in Fig. 1(c). Sulfite, sulfate and nitrate species were observed in this reaction system. Specifically, the bands at 1361 and 1346 $cm^{-1}$ were assigned to aggregated sulfate; bands at 1163 and 1115 $cm^{-1}$ were related to bridging sulfate and bands at 1074 and 1010 $cm^{-1}$ were ascribed to monodentate sulfite (Liu et al., 2012;Yang et al., 2017;Yang et al., 2018). The other bands in the 1620-1370 and 1300-1240 $cm^{-1}$ regions were due to nitrate species, including bridging nitrate (1611, 1246 $cm^{-1}$), bidentate nitrate (1584, 1284 $cm^{-1}$) and monodentate nitrate (1503, 1453 $cm^{-1}$) (Goodman et al., 2001b;Ma et al., 2010). The consumption of OH groups (negative peaks at 3691 and 3630 $cm^{-1}$) and formation of water (3310, 3191, and 3341 $cm^{-1}$) on the particle surface were also observed. These results indicated that $SO_2$ can be partially oxidized to sulfate in the presence of $NO_2$ under dark conditions, which is consistent with previous studies (Ma et al., 2008;Liu et al., 2012), in spite of much lower concentration levels of $SO_2$ and $NO_2$ being used in this study.

The spectra of $TiO_2$ exposed to 200 ppb $SO_2$ and 200 ppb $NO_2$ simultaneously with UV-Vis irradiation were recorded and shown in Fig. 1(d). The bands at 1629, 1584, and 1503 $cm^{-1}$ were related to nitrate species while the bands at 1344, 1284 $cm^{-1}$ and 1177, 1141 $cm^{-1}$ were associated with sulfate species. Compared to the dark experiment of $SO_2$ and $NO_2$ in Fig 1(c), more sulfate species were generated with UV-Vis irradiation, which might be due to the fact that UV-Vis irradiation significantly promotes sulfate formation by generating additional active species (Shang et al., 2010a;Chen et al., 2012) as in the reaction of $SO_2$ alone.

### 3.1.3 Heterogeneous reaction of $SO_2$ and $C_3H_6$ on $TiO_2$

To investigate the heterogeneous reaction with the coexistence of inorganic and organic gases on $TiO_2$, propene was chosen as a representative volatile organic compound, and its effect on the heterogeneous oxidation of $SO_2$ was studied. Under dark conditions, the *in situ* spectra after introduction of 200 ppb $SO_2$+200 ppb $C_3H_6$ were recorded and are shown in Fig. 1(e). No distinguishable products were observed except for the bands at 1074 and 1048 $cm^{-1}$, which were assigned to monodentate sulfite. Compared to the reaction of $SO_2$ alone, the coexistence of $C_3H_6$ had no apparent effect in this dark experiment. With UV-Vis irradiation, the sulfate bands between 1360-1100 $cm^{-1}$ with peaks at 1343, 1289, 1244, 1177 and 1139 $cm^{-1}$ increased with reaction time, as shown in Fig. 1(f). Compared to the reaction of $SO_2$ alone with UV-Vis irradiation, similar spectra were obtained for the $SO_2$+$C_3H_6$ reaction, but the intensities decreased.

### 3.1.4 Heterogeneous reaction of $SO_2$, $NO_2$ and $C_3H_6$ on $TiO_2$

In order approximate the complexity of the real atmosphere, we investigated the heterogeneous reaction of $SO_2$, $NO_2$ and $C_3H_6$ on $TiO_2$. Fig. 1(g) and 1(h) show the dynamic changes of the spectra after introducing these three gases together on $TiO_2$ under dark conditions and with UV-Vis irradiation, respectively. The concentrations of $SO_2$, $NO_2$ and $C_3H_6$ were all 200 ppb. The

reaction of $SO_2/NO_2/C_3H_6$ on $TiO_2$ included both the $SO_2/NO_2$ reaction (Fig. 1(c) and 1(d)) and the $SO_2/C_3H_6$ reaction (Fig. 1(e) and 1(f)) under dark conditions and with UV-Vis irradiation, respectively. Thus, the products included sulfite, nitrate, and some sulfate under dark conditions, while mainly sulfate and nitrate with UV-Vis irradiation.

### 3.2 Sulfate formation and the influence of $NO_2$ and $C_3H_6$

To obtain the area of an individual band for quantitative analysis, a curve-fitting procedure was used employing Lorenz and

Gaussian curves based on the second-derivative spectrum to deconvolute overlapping bands. An example of the analysis for the bands in Fig. 1(b), with a correlation coefficient of 0.992, is shown in Fig. S2 in the Supplemental Information. The band at 1070 is attributed to sulfite, while the bands at 1140, 1178, 1240, 1292 and 1346 $cm^{-1}$ are attributed to sulfate. To avoid interference by nitrate species and other surface products in reactions with the presence of $NO_2$, the peaks at 1198-1135 $cm^{-1}$ were chosen for calculation of the sulfate K-M integrated area.

15        The K-M integrated areas of bridging sulfate in the four reaction systems: (1) $SO_2$; (2) $SO_2+C_3H_6$; (3) $SO_2+NO_2$; (4) $SO_2+NO_2+C_3H_6$ in the dark and with UV-Vis light are shown in Fig. 2(a) and Fig. 2(b), respectively. In the dark experiments, no apparent sulfate was generated in the reaction of $SO_2$ alone. The presence of $C_3H_6$ had no discernible effect on the formation of sulfate in dark experiments. The presence of $NO_2$ promoted the oxidation of $SO_2$ on $TiO_2$, with the result that mostly sulfate was yielded from the reaction of $SO_2+NO_2$. The presence of $NO_2$ seemed to induce the generation of some ROS, which oxidize

S(IV) to S(VI) on $TiO_2$ (Ma et al., 2008;Liu et al., 2012;Ma et al., 2017). The detailed mechanism for this effect has not been fully explored and will be discussed later. It has also been proposed that aqueous oxidation of $SO_2$ by $NO_2$ (as an oxidizing agent) contributed to significant sulfate formation in haze events (Wang et al., 2016b;Cheng et al., 2016). This reaction should not be the main pathway in the reaction systems in this study since the experiments were carried out under dry conditions (RH<1%), although water can still exist, as we mentioned earlier. When $SO_2$ was introduced into the cell with $NO_2$ and $C_3H_6$

together, sulfate formation was less than that in the reaction of $SO_2+NO_2$, probably due to the competition between $SO_2$ and $C_3H_6$ for the ROS due to $NO_2$. In the UV-Vis irradiation experiments, on the contrary, both $NO_2$ and $C_3H_6$ had a distinct suppressing effect on the sulfate formation compared to the individual reaction of $SO_2$. The opposite effect of $NO_2$ on sulfate formation relative to dark experiments may be explained by the different influence of $NO_2$ on the oxidation capacity in the heterogeneous photooxidation, compared to dark experiments. In dark experiments, the contribution of $NO_2$ to the oxidation

capacity is predominant due to the limited availability of ROS, while it becomes of lesser importance when surface ROS are continuously generated in the experiments with UV-Vis irradiation. What's more, the nitrate formation from oxidation of $NO_2$ might block some surface reactive sites, and therefore, resulted in less sulfate formation in the reaction of $SO_2+NO_2$ than that

of $SO_2$ alone with UV-Vis irradiation. To further probe and analyze the total amounts of sulfate in different systems, the samples after reaction in the different experiments were also analyzed by IC. The results, which are shown in Fig. 3, are consistent with the results derived from integrated peak areas in Fig. 2. Since formaldehyde was added to inhibit the oxidation of sulfite to sulfate in the solution, there is a possibility that HMS would be generated in the solution and be measured as sulfate (Moch et al., 2018). However, the possible interference by HMS in the measurement of sulfate by IC will not influence our conclusions on the effects of $NO_2$ and $C_3H_6$, since the K-M integrated area of sulfate in the *In situ* DRIFTS spectra were also compared. Despite the different yields of sulfate under different atmospheres, the presence of UV-Vis irradiation always increased sulfate formation significantly. We also observed that the promotion effect of UV-Vis irradiation on the heterogeneous oxidation of $SO_2$ was most significant for the individual reaction of $SO_2$, while it became less noticeable under more complex pollution, i.e. in the presence of $NO_2$ and some VOCs.

### 3.3 Step-by-step experiments with UV-Vis irradiation and related mechanisms

In the step-by-step experiments, the spectra for $TiO_2$ exposure to 200 ppm $NO_2$ after the first step are shown by the black lines in Fig. 4(a). The nitrate bands at 1611, 1586, 1507, and 1288, 1241 $cm^{-1}$ increased in intensity. When the $NO_2$ was cut off, the particles were purged with air for 1 h, and the spectrum was recorded as the blue line in Fig. 4(a). Air purging did not noticeably change the spectra, except that the nitrate band at 1611 $cm^{-1}$ shifted to 1637 $cm^{-1}$ due to the absorption of water (Ma et al., 2010), indicating a relatively steady adsorption of nitrate species. Then the $NO_2$-preadsorbed $TiO_2$ particles were exposed to $SO_2$ in the third step, marked by red lines in Fig. 4(a). A new band at 1168 $cm^{-1}$ assigned to sulfate appeared and the bands at 1350-1200 $cm^{-1}$ became broader due to the formation of sulfate. Meanwhile, the nitrate bands at 1586 and 1507 $cm^{-1}$ decreased in intensity and even disappeared. The possible reason might be either the replacement of nitrite by sulfate from $SO_2$ heterogeneous photooxidation (Park et al., 2017) or the photolysis of nitrate (Ye et al., 2017).

In the 200 ppm $C_3H_6$ pre-saturated experiment, which is shown in Fig. 4(b), after $C_3H_6$ was introduced into the reaction cell for 2 h, intense bands at 1582, 1541, 1452, 1379, and 1361 $cm^{-1}$ were observed. These principal bands are assigned to carboxylate (-COO, 1582, 1541 $cm^{-1}$) methyl (-$CH_3$, 1452, 1379 $cm^{-1}$), and methyne (-CH, 1361 $cm^{-1}$), respectively (Busca et al., 1987;Idriss et al., 1995). Based on the above bands, the main products could be deemed to be formate and acetate species. After stopping the flow of $C_3H_6$ and flushing the cell with synthetic air for 1 h, the band areas of surface products were reduced, indicating that these species from $C_3H_6$ were not stable and could be removed easily from the surface. The subsequent introduction of $SO_2$ into the system resulted in sulfate formation, as seen by the bands in the 1380-1050 $cm^{-1}$ region. Introducing $NO_2$ and $C_3H_6$ together before $SO_2$ resulted in both nitrate and organic species on $TiO_2$, as shown in Fig. 4(c). It is interesting that some distinct new bands were observed when the surface was exposed to $NO_2+C_3H_6$, such as the bands at 1750, 1682, and 1524 $cm^{-1}$, which could be assigned to $CH_2O$ (Liao et al., 2001), $HNO_3$ (Goodman et al., 2001b) and COO groups (Mattsson and Österlund, 2010), respectively. This may indicate some interaction between $NO_2$ and $C_3H_6$ and a possible influence of $C_3H_6$ on nitrate formation, as well as $NO_2$ on $C_3H_6$ oxidation in the heterogeneous photooxidation.

Figure 5 compares the K-M integrated areas of bridging sulfate (1168 cm$^{-1}$) formed during these step-by-step experiments under different conditions. Compared to the reaction with $SO_2$ alone, the pre-adsorption of $C_3H_6$ on $TiO_2$ did not have any apparent influence. This is consistent with the supposition that the formate and acetate species from heterogeneous oxidation of $C_3H_6$ might be easily removed from the surface. Since introducing $C_3H_6$ with $SO_2$ together suppressed sulfate formation in the heterogeneous photooxidation while pre-adsorption of $C_3H_6$ had little influence, $C_3H_6$ is proposed to compete with $SO_2$ for ROS rather than surface reactive sites in the heterogeneous photooxidation. Instead, the pre-adsorption of $NO_2$ on $TiO_2$ suppressed the formation of sulfate, which might have resulted from the different absorption status of the oxidation products of $NO_2$ and $C_3H_6$. Compared to the experiment introducing $NO_2$ and $SO_2$ simultaneously, sulfate formation was more inhibited with pre-adsorption of $NO_2$ in the first hour, while sulfate formation in these two cases became similar after 1.5 h duration. This may indicate that $NO_2$ suppressed sulfate formation, mainly due to the competition between $SO_2$ and $NO_2$ for surface reactive sites. Compared to the individual reaction of $SO_2$, both pre-adsorption of $NO_2$ and introducing $NO_2$ simultaneously suppressed sulfate formation from the beginning of the heterogeneous photooxidation. It is interesting that pre-adsorption with $NO_2 + C_3H_6$ resulted in much less sulfate formation compared to the pre-adsorption of $NO_2$ or $C_3H_6$, as well as the reaction of $SO_2+NO_2+C_3H_6$. Although the detailed reason for this phenomenon was not discovered in this study, a possible reason might be that the oxidation products from $NO_2$ and $C_3H_6$ blocked some reactive sites on $TiO_2$ and suppressed sulfate formation in heterogeneous photooxidation, since $NO_2$ and $C_3H_6$ were cut off after pre-adsorption and ROS were expected to be generated on $TiO_2$ with UV-Vis irradiation. According to the DRIFTS spectra in Fig. 4(c), besides nitrate, aldehydes (1750 cm$^{-1}$) and carboxylic acids (1524 cm$^{-1}$) were also observed on $TiO_2$ after pre-adsorption with $NO_2 + C_3H_6$.

## 4 Discussion

### 4.1 Dark reactions

The heterogeneous oxidation of $SO_2$ on $TiO_2$ has been investigated by many previous studies. The following mechanisms for $SO_2$ adsorption on $TiO_2$ surfaces have been proposed in previous studies (Nanayakkara et al., 2012):

$$Ti - OH + SO_2 \rightarrow Ti - OSO_2H \qquad (1)$$

$$2Ti - OH + SO_2 \rightarrow Ti_2 - SO_3 \cdot H_2O \qquad (2)$$

$$Ti - O^{2-} + SO_2 \rightarrow Ti - SO_3^{2-} \qquad (3)$$

These adsorption processes result in the conversion of $SO_2$ to sulfite (S(IV)) on the surface. It has been demonstrated that coexisting $NO_2$ can induce the generation of some ROS, which oxidize S(IV) to S(VI) on mineral oxides (Ma et al., 2008;Liu et al., 2012;Ma et al., 2017). There were several possible responsible ROS proposed in previous studies, although the detailed mechanism has not yet been fully explored. One possible ROS is $N_2O_4$, which can undergo hydrolysis to N(III) and N(V) species (Liu et al., 2012;Finlayson-Pitts et al., 2003;Li et al., 2018). These reactive nitrogen species can oxidize S(IV) to S(VI) (Wang et al., 2016b;Li et al., 2018).

$$2Ti - NO_2 \rightarrow Ti_2 - N_2O_4 \qquad (4)$$

$$N_2O_4(ad) \rightarrow NO^+NO_3^- \xrightarrow{H_2O} HNO_3 + HONO \qquad (5)$$

Besides $N_2O_4$, $NO_2$ may also react directly with surface OH and form $HNO_3$ on $TiO_2$ (Liu et al., 2017a). The $HNO_3$ generated through this pathway may also contribute to the oxidation of S(IV) to S(VI). It has also been proposed that aqueous oxidation of $SO_2$ by $NO_2$ (as an oxidizing agent) contributed to significant sulfate formation in haze events (Wang et al., 2016b;Cheng et al., 2016). This aqueous reaction should not be significant in the reaction systems of this study due to the limited amount of water under low RH condition (<1% RH).

When $C_3H_6$ was introduced together with $NO_2$, sulfate formation was less than that in the reaction of $SO_2+NO_2$, probably due to the reaction between $C_3H_6$ and the reactive nitrogen species. The detailed mechanism was not explored in this study. The following reactions may take place in this process.

$$2NO^+NO_3^- + Ti-C_3H_6 \rightarrow H_3CCHO + HCHO + 2NO^+NO_2^- \qquad (6)$$

$$NO^+NO_3^- + HCHO \rightarrow HCOOH + NO^+NO_2^- \qquad (7)$$

$$NO^+NO_3^- + H_3CCHO \rightarrow H_3CCOOH + NO^+NO_2^- \qquad (8)$$

Heterogeneous reactions between $NO_2$ and organics can also lead to nitro-organics on hexane soot (Kwamena and Abbatt, 2008;Al-Abadleh and Grassian, 2000) , which may also occur on the surface of $TiO_2$, and these products blocked some reactive sites for sulfate formation.

## 4.2 Light reactions

With UV illumination, $TiO_2$ can be excited by UV light ($\lambda< 387$ nm), then the photogenerated electrons and holes can react with $H_2O$ and $O_2$ to produce additional ROS (primarily $\cdot O_2^-$ and $\cdot OH$), and oxidize more $SO_2$ to sulfate on $TiO_2$ than that produced under dark conditions (Shang et al., 2010a;Chen et al., 2012).The detailed mechanism was summarized by Chen et al. (Chen et al., 2012) and references therein:

$$TiO_2 + h\nu(\lambda < 387 \text{ nm}) \rightarrow e^-h^+ \rightarrow e^- + h^+ \qquad (9)$$

$$O_2 + e^- \rightarrow \cdot O_2^- \qquad (10)$$

$$H_2O + h^+ \rightarrow \cdot OH + H^+ \qquad (11)$$

Then the $SO_2$ can react with these ROS and promote the formation of sulfate (Shang et al., 2010b):

$$Ti-SO_2 + \cdot O_2^- \rightarrow Ti-SO_3 + O^- \qquad (12)$$

$$Ti-SO_3 + H_2O \rightarrow Ti-H_2SO_4 \qquad (13)$$

$$Ti-SO_3^{2-} + 2\cdot OH \rightarrow Ti-SO_4^{2-} + H_2O \qquad (14)$$

In the UV-Vis irradiation experiments, $NO_2$ had a distinct suppressing effect on the sulfate formation compared to the individual reaction of $SO_2$. Rather than resulting in ROS formation and oxidation of S(IV) to S (VI) in dark experiments, the main reaction of $NO_2$ with the surface ROS resulted in nitrate and nitrite formation in experiments with UV-Vis irradiation (Ndour et al., 2008;Yu and Jang, 2018).

$$Ti - NO_2 + \cdot OH \rightarrow Ti - HONO_2 \tag{15}$$

$$Ti - NO_2 + \cdot O_2^- \rightarrow Ti - NO_2^- + O_2 \tag{16}$$

The nitrate or nitrite generated from the oxidation of $NO_2$ might block some surface reactive sites, since in the step-to-step experiments, the pre-adsorption of $NO_2$ on $TiO_2$ also suppressed the formation of sulfate and resulted in similar sulfate formation to that in the experiment introducing $NO_2$ and $SO_2$ simultaneously. The competition between $SO_2$ and $NO_2$ for surface reactive sites might be the main reason for the fact that the coexistence of $NO_2$ with $SO_2$ resulted in decreased sulfate formation with UV-Vis irradiation in this study. Although Gen et al. (Gen et al., 2019) found that photolysis of nitrate enhanced sulfate formation in wet aerosols, this mechanism may not be applied in this study since the reaction system is quite different from their study. The ROS, which oxidize S(IV) to S(VI), are mainly $\cdot O_2^-$ and $\cdot OH$ in the presence of UV-Vis irradiation rather than the photolysis of nitrate.

$C_3H_6$ also had a distinct suppressing effect on sulfate formation. Similar to $NO_2$, $C_3H_6$ will react with surface ROS.

$$C_3H_6 \xrightarrow{\cdot OH} RCHO \xrightarrow{\cdot OH} RCOOH \xrightarrow{\cdot OH} CO_2 + H_2O \tag{17}$$

where R represents H or an alkyl group. These gaseous products in the photo-oxidation of $C_3H_6$ do not seem to block surface reactive sites, which can explain why the pre-adsorption of $C_3H_6$ on $TiO_2$ did not show an obvious suppressing effect on the formation of sulfate in the step-by-step experiment.

When $C_3H_6 + NO_2$ were introduced simultaneously into the reaction system together with $SO_2$, both competed for ROS with $SO_2$ and therefore resulted in the lowest formation of sulfate among the heterogeneous reactions. Besides, in the step-by-step experiments, the pre-adsorption of $C_3H_6 + NO_2$ on $TiO_2$ suppressed sulfate formation significantly, which indicated that lots of reactive sites for $SO_2$ oxidation might be blocked by these oxidation products in pre-adsorption with UV-Vis irradiation. Karagulian et al. (Karagulian et al., 2009) found that nitrite can induce the photo-oxidation of VOCs on airborne particles and produce organic nitrates and carbonyl compounds. Thus, the formation of organic nitrates may be an important factor to suppress the formation of sulfate due to the blocking effect.

**5 Conclusions and environmental implications**

Based on the experimental results obtained in this study, we propose the following possible mechanisms for the reaction of $SO_2$ in the presence of $NO_2$ and $C_3H_6$ under conditions close to those in the real atmosphere. Under dark conditions at 303 K, $SO_2$ could hardly react on the particle surface and only a few sulfite-like species formed. With reaction time increasing, the adsorption sites on the surface became saturated with sulfite and prevented $SO_2$ from adsorbing on the particles further. Coexisting $NO_2$ could enhance the heterogeneous formation of sulfate with much lower concentrations (200 ppb) relative to previous studies (~100 ppm) (Ma et al., 2008;Liu et al., 2012;Zhao et al., 2018). The presence of $C_3H_6$ had little effect on sulfate formation in the heterogeneous reaction of $SO_2$ but suppressed sulfate formation in the heterogeneous reaction of $SO_2$ and $NO_2$, because $C_3H_6$ could react ROS generated in the adsorption of $NO_2$. When irradiation was introduced into the system, the ROS such as $\cdot OH$ and $\cdot O_2^-$ could initiate photocatalytic oxidation of S(IV) species to sulfate. Sulfate formation was

suppressed significantly with the coexistence of $NO_2$ and/or $C_3H_6$ in the presence of UV-Vis light. The formation of nitrate, carbonyl compounds, and organic nitrate consumed both available ROS and surface reactive sites.

These results indicated that heterogeneous oxidation of $SO_2$ might be influenced by the co-existing inorganic and organic gas pollutants under complex pollution conditions due to the competition for ROS and active surface sites among them.

In this study, only one VOC was investigated, while the heterogeneous oxidation of various VOCs has been reported in previous studies (Niu et al., 2017;Du et al., 2000). When a VOC and $SO_2$ coexist, the competition for ROS and surface reactive sites between the VOC and $SO_2$ is likely to suppress sulfate formation in the heterogeneous reactions, such as that observed for the presence of $CH_3CHO$ on $\alpha$-$Fe_2O_3$ in dark experiments (Zhao et al., 2015), the presence of $C_7H_{16}$ on $TiO_2$ with UV-Vis irradiation (Du et al., 2000), and the presence of $C_3H_6$ on $TiO_2$ under dark condition or with UV-Vis irradiation in this study.

Due to the different properties of the oxidation products, the influence of coexisting VOCs might be different for different VOC species and on different mineral dusts. Some coexisting VOCs, such as HCOOH on $\alpha$-$Fe_2O_3$ (Wu et al., 2013), and HCHO in aerosol water (Moch et al., 2018;Song et al., 2019) might enhance sulfate formation. These results highlighted the very complex heterogeneous reaction processes that take place under complex air pollution conditions due to the ubiquitous interactions between organic and inorganic species. For better estimation of heterogeneous sulfate formation, the kinetics of

the heterogeneous oxidation of $SO_2$ must be developed with consideration of the influence of coexisting VOCs and other inorganic gases.

### Data availability

All the data related to this paper may be requested from the corresponding author: qxma@rcees.ac.cn

### Author contribution

QM, BC and HH designed the study. YW, WY and BC carried out the experiments. BC, WY, JM, and QM analyzed the data with input from all co-authors. BC and YW wrote the paper with contribution from YL, JM, WY, and PZ on the editing of the paper.

### Competing interests

The authors declare that they have no conflict of interest.

### Acknowledgements

This work was supported by the National Key R&D Program of China (2018YFC0506901), National Natural Science Foundation of China (41877304, 21876185, 91744205), the National Research Program for Key Issues in Air Pollution Control (DQGG0103), and the Youth Innovation Promotion Association, CAS (2018060, 2018055, 2017064).

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

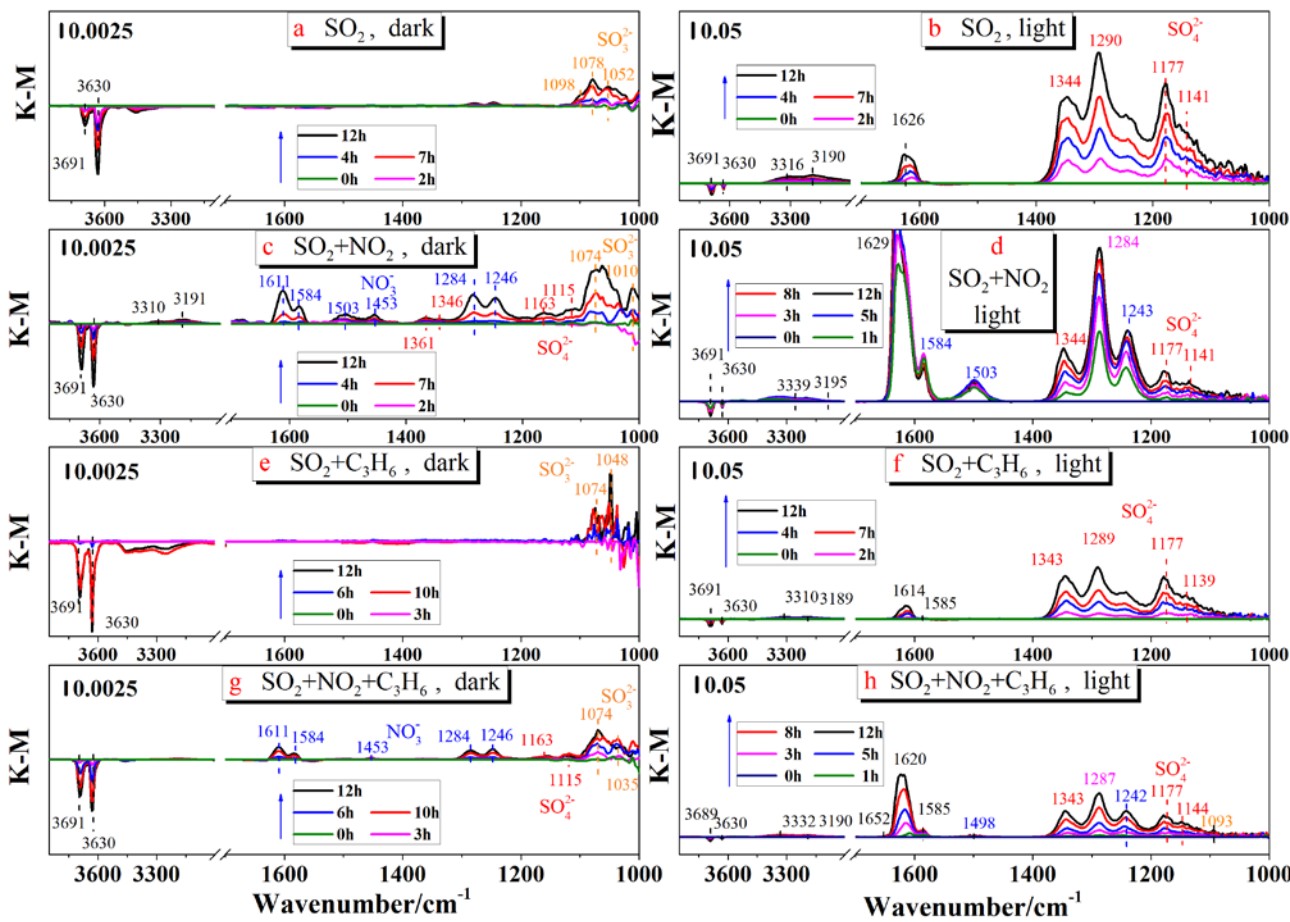

**Figure 1: Dynamic changes in the *in situ* DRIFTS spectra of the TiO₂ sample as a function of time at 303K in a flow of 20% O₂ + 80% N₂ with 200 ppb SO₂ under dark conditions (a) and with UV-Vis light (b); with 200 ppb SO₂ + 200 ppb NO₂ under dark conditions (c) or with UV-Vis light (d); with 200 ppb SO₂ + 200 ppb C₃H₆ under dark conditions (e) or with UV-Vis light (f); with 200 ppb SO₂ + 200 ppb NO₂+ 200 ppb C₃H₆ + under dark conditions (g) or with UV-Vis light (h).**

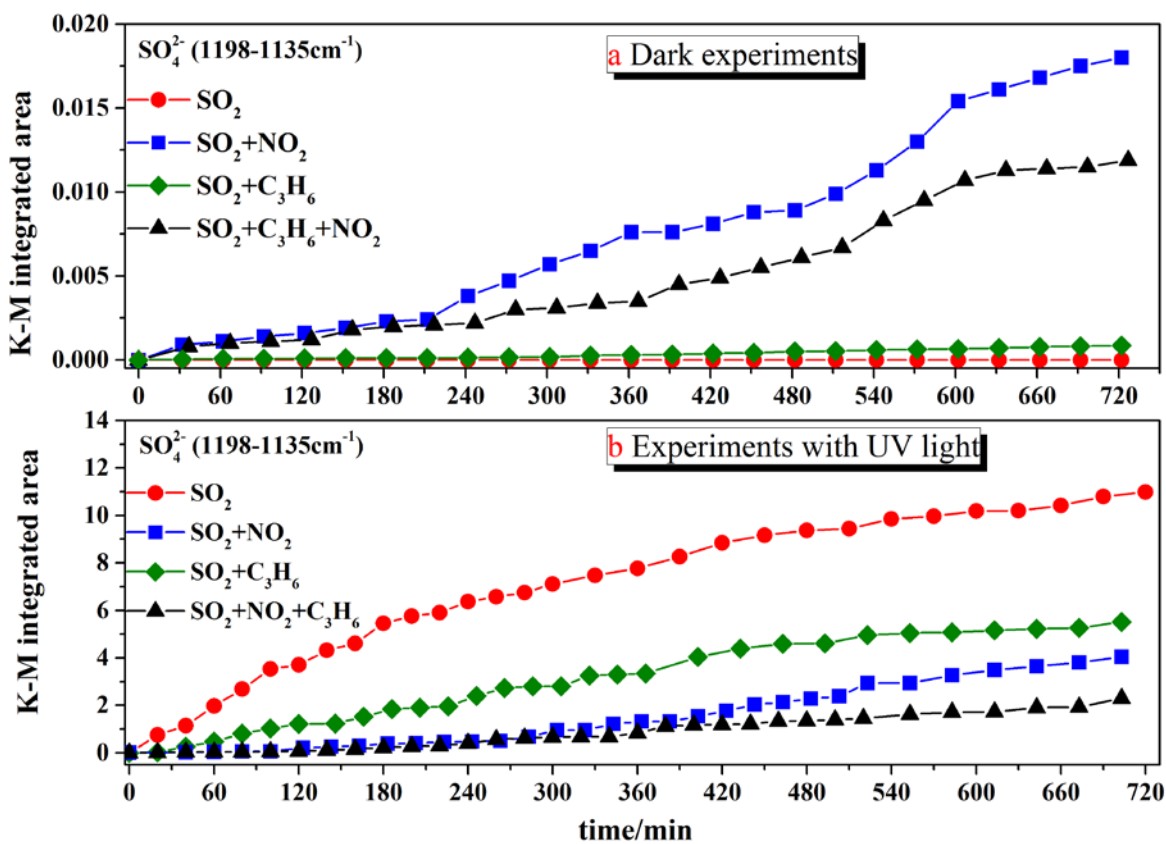

**Figure 2: Integrated absorbance of the sulfate band (1198-1135 cm⁻¹) observed during the reaction of 200 ppb SO₂, 200 ppb SO₂+200 ppb NO₂, 200 ppb SO₂+200 ppb C₃H₆, 200 ppb SO₂+200 ppb NO₂+200 ppb C₃H₆ in dark experiments (a) and experiments with UV-Vis light (b).**

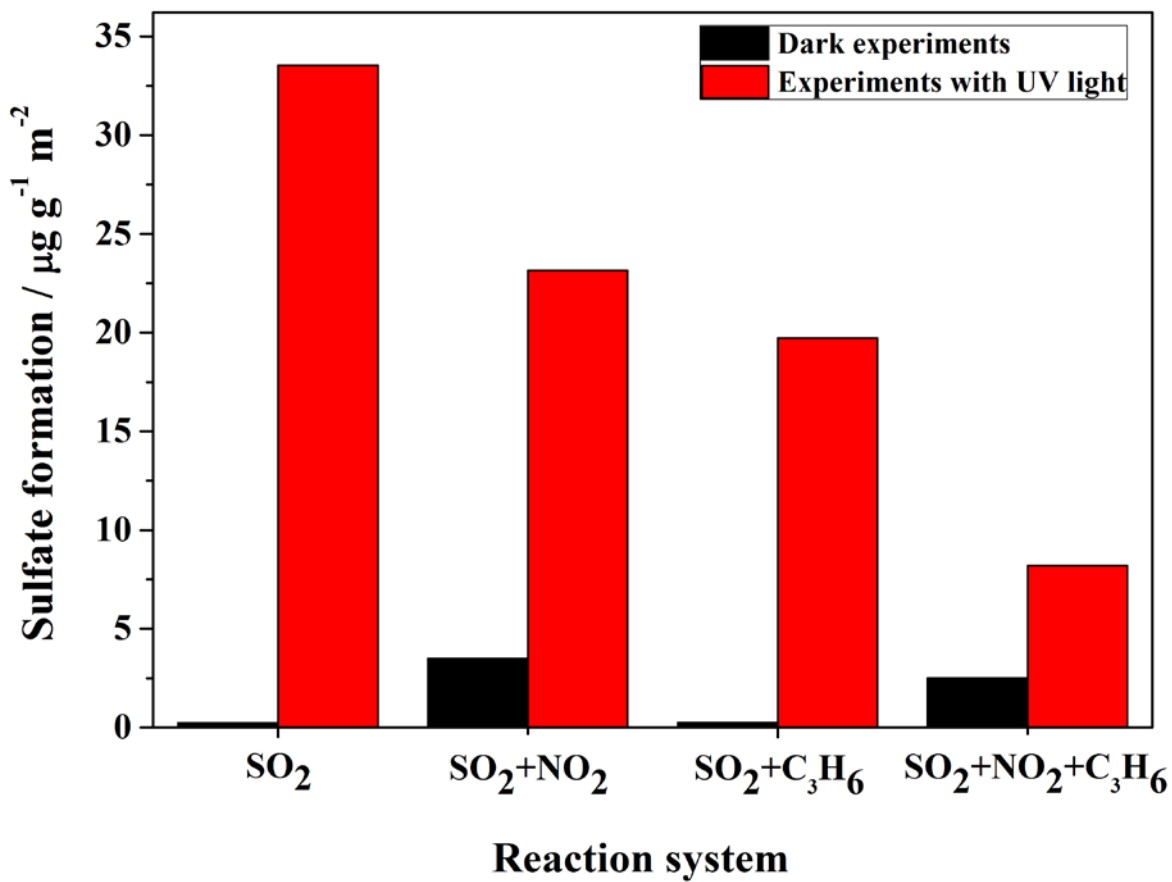

**Figure 3: Ion chromatography results of the amounts of sulfate (product per unit mass/surface area of sample) formed on the surface of TiO₂ after reaction with SO₂, SO₂+NO₂, SO₂+C₃H₆ and SO₂+C₃H₆+NO₂ in experiments under dark conditions or with UV-Vis light. Since formaldehyde was added to inhibit the oxidation of sulfite to sulfate in the solution, there is a possibility that HMS would be generated in the solution and be measured as sulfate.**

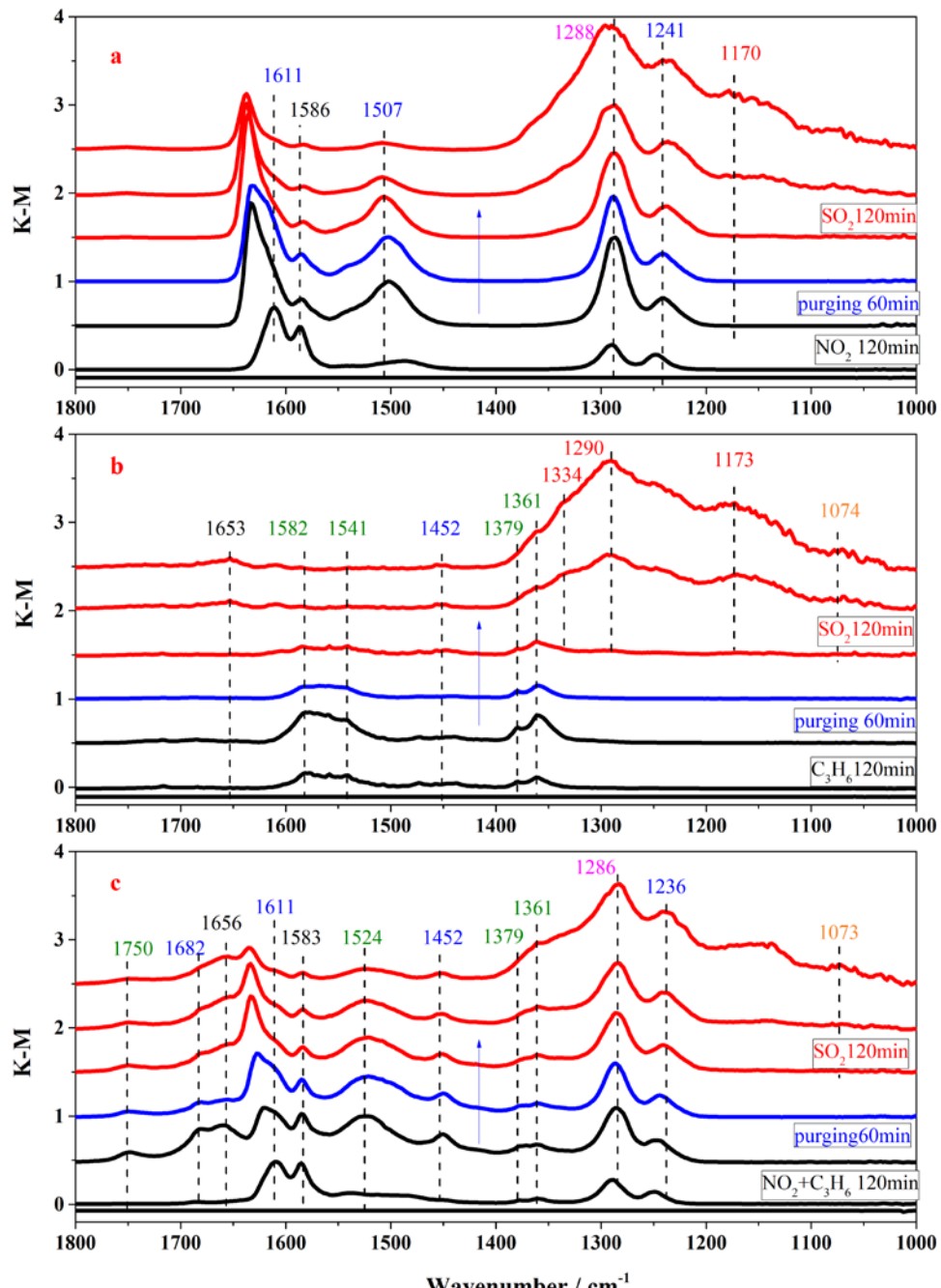

**Figure 4:** *In situ* **DRIFTS spectra of surface products on TiO₂ in the step-by-step exposure experiments with irradiation: (a) exposure to 200 ppm NO₂ for 2 h (black lines), after purging 1 h (blue line), and then to 200 ppm SO₂ for 2 h (red lines); (b) exposure to 200 ppm C₃H₆ for 2 h (black lines), after purging 1 h (blue line), and then to 200 ppm SO₂ for 2 h (red lines); (c) exposure to 200 ppm NO₂+200 ppm C₃H₆ for 2 h (black lines), after purging 1 h (blue line), and then to 200 ppm SO₂ for 2 h (red lines).**

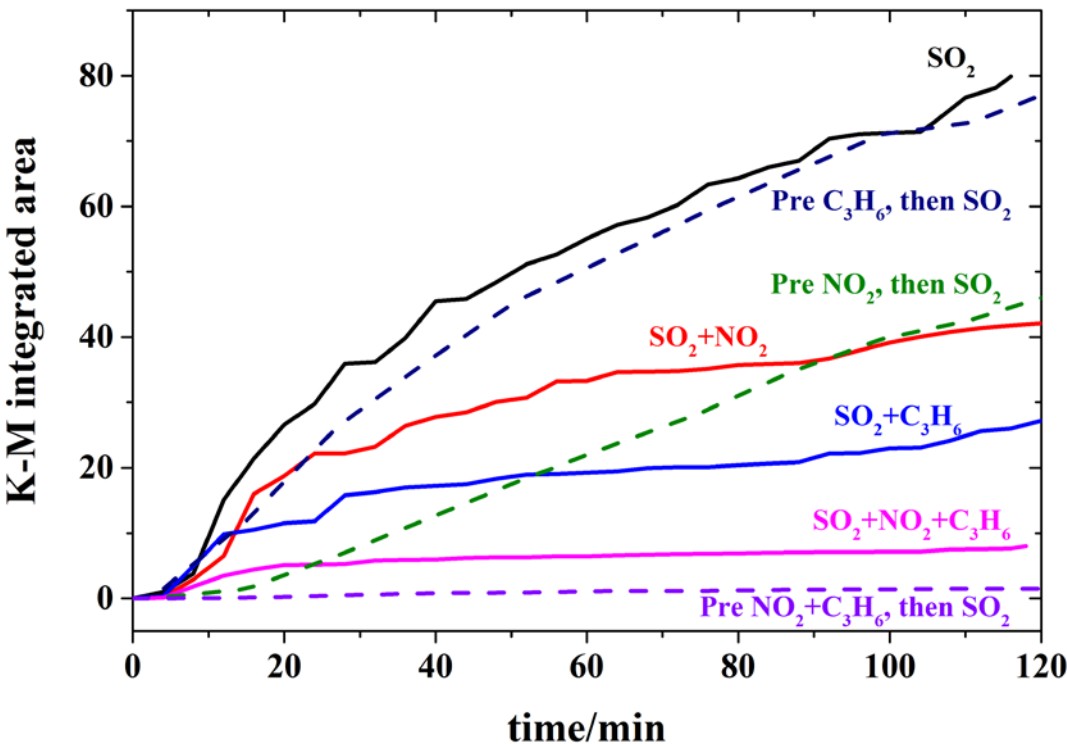

**Figure 5: Integrated absorbance of the sulfate band (1168 cm$^{-1}$) for the illuminated reactions with UV-Vis light of 200 ppm SO$_2$ (black, solid), 200 ppm SO$_2$ on a 200 ppm C$_3$H$_6$-presaturated surface (blue, dashed), 200 ppm SO$_2$+200 ppm NO$_2$ (red, solid), 200 ppm SO$_2$ on a 200 ppm NO$_2$-presaturated surface (green, dashed), 200 ppm SO$_2$+200 ppm C$_3$H$_6$ (blue, solid), 200 ppm SO$_2$+200 ppm NO$_2$+200 ppm C$_3$H$_6$ (pink, solid), and 200 ppm SO$_2$ on a 200 ppm NO$_2$+200 ppm C$_3$H$_6$-presaturated surface (purple, dashed).**

**Table 1: Vibrational frequencies of chemisorbed species formed on TiO₂.**

| surface species | | frequencies(cm$^{-1}$) | References |
|---|---|---|---|
| $SO_3^{2-}/HSO_3^-$ | monodentate sulfite | 1098  1078  1052 | (Liu et al., 2012;Nanayakkara et al., 2012) |
| $SO_4^{2-}$ | state of aggregation | 1344 | (Nanayakkara et al., 2012) |
| | bidentate | 1290 | (Yang et al., 2005) |
| | bridging | 1177  1141 | (Chen et al., 2007) |
| $NO_3^-$ | bridging | 1611  1246 | (Goodman et al., 2001a;Underwood et al., 1999;Hadjiivanov and Knözinger, 2000) |
| | bidentate | 1584  1284 | (Hadjiivanov and Knözinger, 2000) |
| | monodentate | 1503  1453 | (Piazzesi et al., 2006) |
| $HNO_3$ | | 1682 | (Goodman et al., 2001b) |
| $COO^-$ | | 1585  1541 | (Busca et al., 1987;Idriss et al., 1995;Rachmady and Vannice, 2002a;Mattsson and Österlund, 2010) |
| $-CH_3$ | | 1452  1379 | (Busca et al., 1987) |
| -CH | | 1361 | (Rachmady and Vannice, 2002b) |
| -CHO | | 1745 | (Liao et al., 2001) |
| $H_2O$ | bending vibration | 1626 | (Goodman et al., 1999) |
| OH | isolated bicoordinated (on Ti atoms) | 3690 | (Primet et al., 1971) |
| | H-bonded | 3631 | (Tsyganenko and Filimonov, 1973;Ferretto and Glisenti, 2003) |
| OH | adsorbed water | 3456  3310  3190 | (Tarbuck and Richmond, 2006) |