# Peer review of "Effects of NO2 and C3H6 on the heterogeneous oxidation of SO2 on TiO2 in the presence or absence of UV-Vis irradiation"

_Atmospheric Chemistry and Physics, 2019_

## Referee Comment (RC1) · Anonymous Referee #2 · 30 Jul 2019

The comment was uploaded in the form of a supplement: https://www.atmos-chem-phys-discuss.net/acp-2019-532/acp-2019-532-RC1-supplement.pdf

---

## Referee Comment (RC2) · Anonymous Referee #1 · 9 Aug 2019

This manuscript describes a set of experiments looking at the effects of UV radiation, NO2, and C3H6 on heterogeneous SO2 oxidation. The authors demonstrate that on TiO2 aerosols the presence of NO2 alone or NO2 and C3H6 can suppress sulfate formation when in the presence of UV light. The authors also show that in dark conditions NO2 alone or NO2 and C3H6 can increase heterogeneous sulfate formation, but that the presence of C3H6 decreases heterogeneous sulfate formation compared to heterogeneous sulfate formation with NO2 only. That the presence of VOCs may suppress sulfate formation is an interesting and little focused on point. The authors need to add a bit more of a discussion of why they interpret their experiments as supporting their proposed mechanism. For example, the authors suggest that the presence of

C3H6 inhibited heterogeneous sulfate formation with NO2 by competing with SO2 for reactive oxygen species or active sites on the aerosol. It is not clear to readers how the authors reach this particular conclusion. The authors mention that the presence of NO2 induced the generation of reactive oxygen species, but the mechanism behind this is never satisfactorily explained. It is also unclear if the authors are saying that C3H6 competes with SO2 or with NO2 for active sites on the aerosol. A more detailed discussion of the mechanism behind the 'dark' oxidation of SO2 in the presence of NO2 and identifying the points in that mechanism in which C3H6 interferes could help clarify these issues. There are also some other outstanding issues listed below. Nevertheless, the key point that VOCs may suppress heterogeneous sulfate formation in dark conditions is a very important one. Altogether, the manuscript requires some important revisions before publication in ACP.

Other general comments:

1. The authors need to better explain why TiO2 is a good compound for approximating the heterogeneous oxidation of SO2 on mineral dust aerosols. The authors mention 4 studies using different types of mineral oxides (line 21 page 2). What were the differences between these studies attributable to the different mineral oxide used? Why did the authors in this study choose TiO2 instead of CaO, a-Fe2O3 or MgO, when calcium, magnesium, and iron are usually a much larger portions of mineral dust? What do the authors anticipate the effect of using different mineral oxides would be on their experiments?

2. Similar to the point in comment 1, the authors should elaborate further why propene was selected as a representative VOC. What evidence is there that propene is representative of different VOCs? How might the type of VOC used affect results?

3. As the authors are likely aware, it has also been proposed that a significant sulfate formation pathway for Chinese winter haze is heterogeneous oxidation of SO2 by NO2 (e.g. Wang et al., 2016; Cheng et al., 2016). The authors need to demonstrate that

this reaction is not significant in their experiments. This could be done by showing how NO2 changes along with SO2 in their experiments. In the proposed mechanism of the authors, NO2 acts as a catalyst and therefore concentrations should not change. In the alternative mechanism NO2 is the oxidizing agent and therefore should be depleted along with SO2 as sulfate forms. If it turns out this other reaction is significant, this should be accounted for.

Wang, G., Zhang, R., Gomez, M. E., Yang, L., Zamora, M. L., Hu, M., et al. (2016). Persistent sulfate formation from London Fog to Chinese haze. Proceedings of the National Academy of Sciences, 113(48), 13630–13635. https://doi.org/10.1073/pnas.1616540113

Cheng, Y., Zheng, G., Wei, C., Mu, Q., Zheng, B., Wang, Z., et al. (2016). Reactive nitrogen chemistry in aerosol water as a source of sulfate during haze events in China. Science Advances, 2(12), e1601530. https://doi.org/10.1126/sciadv.1601530

4. The concluding paragraph of the introduction has multiple sentences that are oddly phrased.

5. In the IC section of the methods, what column type was used? Moch et al., 2018 found that certain IC column types could easily separate hydroxymethanesulfonate (HMS) and sulfate and others could not. Since the author's method involves adding a 1% formaldehyde solution to the samples, this would create HMS and possible an artifact in the IC measurements depending on the column type. Additionally, the authors mention that CH2O was observed when the surface was exposed to NO2 and C3H6, which might also indicate HMS formation

Moch, J. M., Dovrou, E., Mickley, L. J., Keutsch, F. N., Cheng, Y., Jacob, D. J., et al. (2018). Contribution of Hydroxymethane Sulfonate to Ambient Particulate Matter: A Potential Explanation for High Particulate Sulfur During Severe Winter Haze in Beijing. Geophysical Research Letters, 45(21), 11,969-11,979. https://doi.org/10.1029/2018GL079309

6. Many parts of the Results and Discussion section are better suited for placement in the methods section (e.g. the first and third sentence of section 3.1.1, large parts of the first paragraph of 3.3, etc.). The authors should consider moving sentences that describe how the experiments were conducted to the methods section and focus only on the results in the results section.

Other comments:

1. Line 12 on page 2 says that "SO2 can be irreversibly converted into sulfite, bisulfite or sulfate." This is incorrect for sulfite and bisulfite. Even if the particular conditions of the particle mean that sulfite or bisulfite are stable, if conditions change the SO2-HSO3–SO32- equilibrium can shift and the authors should therefore avoid the use of the word "irreversibly" as applied to HSO3- and SO32- formation.

2. Line 14 on page 2 says the authors say "low concentrations (200 ppb)." Was this a typo and the authors meant to write ppt? If not and the authors may mean low for a laboratory setting, but this type of phrasing could be confusing to non-laboratory scientists who may be interested in the author's work since atmospheric propene concentrations are rarely more than a couple of ppb. Later the authors say they used pollutants are "close to ambient concentration" (line 28 page 8), but 200 ppb NO2 and SO2 is much higher than ambient concentrations of these pollutants even during the extremely severe winter haze in Beijing. The authors should either include reference values for the concentrations of these gases in the laboratory compared to the atmosphere, or drop the use of "low concentrations" or "ambient" all together.

3. Line 18 on page 2 regarding states "NO2 was proposed to act as a catalyst to activate O2 in the oxidation." This was a bit confusing, but I assume this means that the authors mean NO2 catalyzed the oxidation of SO2 by O2. If that is correct the authors should change the sentence. Since there is also the heterogeneous oxidation of SO2 by NO2, the author be sure to clarify when the mechanisms involving NO2 they are referring to have SO2 oxidized by O2 and catalyzed by NO2 or have SO2 oxidized

by NO2. I believe in most instances the authors are referring to the former reaction (i.e. catalyzed by NO2 and oxidized by O2).

4. With regards to the formation of hydroxymethanesulfonate (line 26-27 page 2), it would be appropriate for authors to also cite Moch et al., 2018 (referenced above) which also proposed the reaction of CH2O and sulfite/bisulfite in northern China winter haze.

---

## Referee Comment (RC3) · Anonymous Referee #3 · 9 Aug 2019

This manuscript presents an experimental study on the influence of NO2 and a specific VOC (propene) on the heterogeneous production of sulfate on TiO2 particles. The study argues for the complexity in the situation of multiple precursors. The topic fits well in the journal. However, there are significant issues within the manuscript. Below are the major, minor and technical comments. They should be satisfactorily addressed before consideration for publication in the final ACP. Major: A major question that I have is on the set up of the experiments in which many details are missing in the current manuscript. Specifically, (1) is relative humidity controlled? A lot of previous studies show the importance of RH in heterogeneous reactions. RH (or the abundance of water vapor) also impacts gas phase reactions through HOx cycle. (2) about UV light

illumination. What is the amplitude and the range of wavelength? Is it represent of the real atmosphere? (3) the detection of ion chromatography. Is it interfered by HMS hydroxymethanesulfonate? (4) Rational of the choice of materials: TiO2 and propene. How well do they represent the aerosol particles and VOCs? These above questions should be clearly answered in the manuscript. The second one is on the structure of the manuscript. Currently, a big chunk of the method description resides in the results and discussion. I suggest that the authors should re-organize the structure and separate method, results, and discussion (three sections). The experiments conducted in this study should be summarized at first in the method section. In the discussion section, a more thorough and clear discussion on the influencing factors of SO2 oxidation should be provided. The third one is on the proposed mechanisms which in my opinion are not well justified. The study intends to explore the underlying mechanisms through different combinations of chemical precursors. The proposed mechanisms are specifically related to the production and/or competition for ROS and surface reactive sites. But the study does not provide a good way in the experiments to argue for the importance of ROS and reactive sites. What are differences in terms of production and fate of ROS under dark and illumination conditions? Is there a way to detecting the saturation of surface reactive sites? Minor: Page 5, Line 9-10: Elaborate on the processes leading to surface water formation. Page 9, Paragraph 2: Elaborate on the different effects of different VOCs from previous studies. The authors may consider move Figure 2 to the supplemental. Technical: Page 1, Line 17: full expression for "DRIFTS" Page 2, Line 5: "the mechanisms of heterogeneous reaction processes as well as their"

---

## Author Comment (AC1) · 17 Oct 2019

We appreciate the comments from the reviewer's on this manuscript. We have answered them point by point in the attached file. The revised manuscripts (with and without revising history) are also attached.

Please also note the supplement to this comment: https://www.atmos-chem-phys-discuss.net/acp-2019-532/acp-2019-532-AC1-supplement.zip

2019.

---

## Author Response (AR2)

**Ms. Ref. No.: acp-2019-532**

**Title: "Effects of NO₂ and C₃H₆ on the heterogeneous oxidation of SO₂ on TiO₂ in the presence or absence of UV irradiation"**

We appreciate the comments from the reviewer's on this manuscript. We have answered them in the following paragraphs (the text in italics is the reviewer comments, followed by our response) point by point. The page and line numbers in the response are from the revised manuscript.

**Response for Reviewer #1**

*This manuscript describes a set of experiments looking at the effects of UV radiation, NO2, and C3H6 on heterogeneous SO2 oxidation. The authors demonstrate that on TiO2 aerosols the presence of NO2 alone or NO2 and C3H6 can suppress sulfate formation when in the presence of UV light. The authors also show that in dark conditions NO2 alone or NO2 and C3H6 can increase heterogeneous sulfate formation, but that the presence of C3H6 decreases heterogeneous sulfate formation compared to heterogeneous sulfate formation with NO2 only. That the presence of VOCs may suppress sulfate formation is an interesting and little focused on point. The authors need to add a bit more of a discussion of why they interpret their experiments as supporting their proposed mechanism. For example, the authors suggest that the presence of C3H6 inhibited heterogeneous sulfate formation with NO2 by competing with SO2 for reactive oxygen species or active sites on the aerosol. It is not clear to readers how the authors reach this particular conclusion. The authors mention that the presence of NO2 induced the generation of reactive oxygen species, but the mechanism behind this is never satisfactorily explained. It is also unclear if the authors are saying that C3H6 competes with SO2 or with NO2 for active sites on the aerosol. A more detailed discussion of the mechanism behind the 'dark' oxidation of SO2 in the presence of NO2 and identifying the points in that mechanism in which C3H6 interferes could help clarify these issues. There are also some other outstanding issues listed below. Nevertheless, the key point that VOCs may suppress heterogeneous sulfate formation in dark conditions is a very important one. Altogether, the manuscript requires some important revisions before publication in ACP.*

**Response:** Thanks for the reviewer's comments. The reviewer mentioned two main aspects about the mechanisms, one is related to that the presence of VOCs may suppress sulfate formation, and the other is for that NO₂ induced the generation of reactive oxygen species. In the revised manuscript, a discussion section about the proposed mechanisms on these effects are added.

      For that NO₂ induced the generation of reactive oxygen species, we added the following discussions.

Page 9, Line 26 - Page 10, Line 6:

"These adsorption processes result in the conversion of $SO_2$ to sulfite (S(IV)) on the surface. It has been demonstrated that coexisting $NO_2$ can induce the generation of some ROS, which oxidize S(IV) to S(VI) on mineral oxides (Ma et al., 2008;Liu et al., 2012;Ma et al., 2017). There were several possible responsible ROS proposed in previous studies, although the detailed mechanism has not yet been fully explored. One possible ROS is $N_2O_4$, which can undergo hydrolysis to N(III) and N(V) species (Liu et al., 2012;Finlayson-Pitts et al., 2003;Li et al., 2018). These reactive nitrogen species can oxidize S(IV) to S(VI) (Wang et al., 2016b;Li et al., 2018).

$$2Ti - NO_2 \rightarrow Ti_2 - N_2O_4 \qquad\qquad (4)$$

$$N_2O_4(ad) \rightarrow NO^+NO_3^- \xrightarrow{H_2O} HNO_3 + HONO \qquad\qquad (5)$$

Besides $N_2O_4$, $NO_2$ may also react directly with surface OH and form $HNO_3$ on $TiO_2$ (Liu et al., 2017a). The $HNO_3$ generated through this pathway may also contribute to the oxidation of S(IV) to S(VI). It has also been proposed that aqueous oxidation of $SO_2$ by $NO_2$ (as an oxidizing agent) contributed to significant sulfate formation in haze events (Wang et al., 2016b;Cheng et al., 2016). This aqueous reaction should not be significant in the reaction systems of this study due to the limited amount of water under low RH condition (<1% RH)."

For that the presence of VOCs may suppress sulfate formation, or that the presence of $C_3H_6$ inhibited heterogeneous sulfate formation with $NO_2$ by competing with $SO_2$ for reactive oxygen species or active sites on the aerosol, we added the following discussions.

Page 10, Line 7-15:

"When $C_3H_6$ was introduced together with $NO_2$, sulfate formation was less than that in the reaction of $SO_2+NO_2$, probably due to the reaction between $C_3H_6$ and the reactive nitrogen species. The detailed mechanism was not explored in this study. The following reactions may take place in this process.

$$2NO^+NO_3^- + Ti - C_3H_6 \rightarrow H_3CCHO + HCHO + 2NO^+NO_2^- \qquad\qquad (6)$$

$$NO^+NO_3^- + HCHO \rightarrow HCOOH + NO^+NO_2^- \qquad\qquad (7)$$

$$NO^+NO_3^- + H_3CCHO \rightarrow H_3CCOOH + NO^+NO_2^- \qquad\qquad (8)$$

Heterogeneous reactions between $NO_2$ and organics can also lead to nitro-organics on hexane soot (Kwamena and Abbatt, 2008;Al-Abadleh and Grassian, 2000) , which may also occur on the surface of $TiO_2$, and these products blocked some reactive sites for sulfate formation."

Similarly, we also added the mechanisms in the light experiments in the revised manuscript.

*Other general comments:*

*1. The authors need to better explain why TiO2 is a good compound for approximating the heterogeneous oxidation of SO2 on mineral dust aerosols. The authors mention 4 studies using different types of mineral oxides (line 21 page 2). What were*

*the differences between these studies attributable to the different mineral oxide used? Why did the authors in this study choose TiO2 instead of CaO, a-Fe2O3 or MgO, when calcium, magnesium, and iron are usually a much larger portions of mineral dust? What do the authors anticipate the effect of using different mineral oxides would be on their experiments?*

**Response:** We have the same concern as the reviewer and plan to investigate these processes on different mineral dust and authentic dust particles in the future. Actually, the heterogeneous reactivity depends greatly on the properties of mineral oxides, such as the acid−base nature, or the redox properties (Tang et al., 2016;Yang et al., 2016;Yang et al., 2019). For example, basic mineral oxides such as MgO and CaO were more active than acidic $SiO_2$ for the heterogeneous reaction of $NO_2$, while $Fe_2O_3$, with its unique $Fe^{2+}/Fe^{3+}$ redox chemistry, favors the formation of $SO_4^{2−}$ and the heterogeneous conversion of $NO_2$. There are many previous studies using different types of mineral oxides and some using authentic dust particles to study heterogeneous process. The uptake coefficient of $SO_2$ onto pure metal oxides is generally larger than authentic dust particles. Semiconductive metal oxides were known to be responsible for heterogeneous photo-oxidation of tracer gases. In this study, with the coexisting of multigas, the oxidation system is relatively complex, and both dark experiments and UV experiments were carried out. In order to better study the effects of $NO_2$ and $C_3H_6$ on the heterogeneous oxidation in the presence or absence of UV irradiation, we chose $TiO_2$ since it is a semiconductor material and a well-known photocatalyst, although it is less abundant than CaO, $Fe_2O_3$ or MgO in the atmosphere. $TiO_2$ particles are expected to provide important surfaces for heterogeneous photocatalysis of atmospheric gases due to their high photocatalytic activity, especially with the growing application of $TiO_2$ in human activities (Chen et al., 2012). We tried to anticipate the effect of choosing different mineral oxides, but it is difficult since very few previous studies compared heterogeneous reactions on different mineral dust, especially for heterogeneous reactivity in multigas coexisting system. For the dark experiments, since NOx enhance $SO_2$ oxidation on different mineral dusts (He et al., 2014), we anticipate the effects of coexisting of $NO_2$ and $C_3H_6$ might be ubiquitous; while for the UV experiments, we anticipate the effects of $NO_2$ and $C_3H_6$ might be  applied to semiconductive metal oxides and authentic dust particles contain semiconductive metal oxides. In the revised manuscript, the following statement were added.

Page 3, Line 13-19:

ADD "In order to better study the effects of $NO_2$ and $C_3H_6$ on the heterogeneous oxidation in a relatively complex oxidation system (with coexistence of multiple gases, in both dark and illuminated conditions), we chose $TiO_2$ due to the fact that it is a semiconductor material and a well-known photocatalyst. $TiO_2$ has been widely reported to be present in airborne particulate matter (PM) (Chen et al., 2012), Although $TiO_2$ represents only a relatively small portion of the mass of PM and is less abundant than CaO, $Fe_2O_3$ or MgO, the $TiO_2$ particles are expected to provide important surfaces for heterogeneous photocatalysis of atmospheric gases due to their high photocatalytic activity, especially with the growing application of $TiO_2$ in human activities (Chen et al., 2012)."

Page 3, Line 22-25:

ADD "However, we must point out that the heterogeneous reactivity depends greatly on the properties of the mineral oxides, such as acid−base nature or redox properties (Tang et al., 2016;Yang et al., 2016;Yang et al., 2019), while different

VOCs may also have quite different heterogeneous and photochemical reactivity. Investigating these processes on different mineral dust and authentic dust particles with different types of VOCs is needed in future studies."

*2. Similar to the point in comment 1, the authors should elaborate further why propene was selected as a representative VOC. What evidence is there that propene is representative of different VOCs? How might the type of VOC used affect results?*

**Response:** Besides the revision as mentioned in the response to comment 1. We further elaborate why propene was selected in the revised manuscript as follows:

      Page 3, Line 19-25:

      ADD "Propene is selected as a representative VOC since it is the most abundant alkene compound in the atmosphere, and coexists with $NO_x$ in vehicle exhaust emission (Wang et al., 2016a). Propene is widely used as an accelerator in photochemical reactions in some smog chamber studies (Jang and Kamens, 2001;Song et al., 2007). The relatively simple oxidation products and well understood oxidation mechanism of propene are also helpful in explaining our experimental results. Propene is selected also due to the high vapor pressure of its oxidation products, which normally do not generate condensed organic aerosol (Odum et al., 1996)."

*3. As the authors are likely aware, it has also been proposed that a significant sulfate formation pathway for Chinese winter haze is heterogeneous oxidation of SO2 by NO2 (e.g. Wang et al., 2016; Cheng et al., 2016). The authors need to demonstrate that this reaction is not significant in their experiments. This could be done by showing how NO2 changes along with SO2 in their experiments. In the proposed mechanism of the authors, NO2 acts as a catalyst and therefore concentrations should not change. In the alternative mechanism NO2 is the oxidizing agent and therefore should be depleted along with SO2 as sulfate forms. If it turns out this other reaction is significant, this should be accounted for.*

*Wang, G., Zhang, R., Gomez, M. E., Yang, L., Zamora, M. L., Hu, M., et al. (2016). Persistent sulfate formation from London Fog to Chinese haze. Proceedings of the National Academy of Sciences, 113(48), 13630–13635. https://doi.org/10.1073/pnas.1616540113*

*Cheng, Y., Zheng, G., Wei, C., Mu, Q., Zheng, B., Wang, Z., et al. (2016). Reactive nitrogen chemistry in aerosol water as a source of sulfate during haze events in China. Science Advances, 2(12), e1601530. https://doi.org/10.1126/sciadv.1601530*

**Response:** Thanks for the reminding. $NO_2$ oxidized $SO_2$ as an oxidizing agent happened under specified conditions, such as in aqueous phase and in the presence of $NH_3$ that the pH is not very low. This reaction should not be significant since our experiments were carried out under dry condition. Since $NO_2$ also transformed to nitrate in the reaction itself and there was competition between $NO_2$ and $SO_2$ for surface active sites, it was not easy to demonstrate the role of $NO_2$ from how it changes.

      Page 7, Line 21-24:

ADD "It has also been proposed that aqueous oxidation of $SO_2$ by $NO_2$ (as an oxidizing agent) contributed to significant sulfate formation in haze events (Wang et al., 2016b;Cheng et al., 2016). This reaction should not be the main pathway in the reaction systems in this study since the experiments were carried out under dry conditions (RH<1%), although water can still exist, as we mentioned earlier."

*4. The concluding paragraph of the introduction has multiple sentences that are oddly phrased.*

**Response:** The paragraph were rewritten in the revised version. Besides the added sentences, some original sentence were modified:

Page 3, Line 9-13:

"In spite of these studies involving the heterogeneous oxidation of $SO_2$ under various conditions, it is not fully understood how the heterogeneous oxidation of $SO_2$ is influenced by co-existing pollutants under dark or illumination conditions. Meanwhile, the interactions between organic and inorganic species in the heterogeneous oxidation of $SO_2$ at low concentrations have not been deeply researched yet. In this study, we focus on the effects of co-existing $NO_2$ and propene at low concentrations (200 ppb) on the heterogeneous oxidation of $SO_2$ on $TiO_2$ with *in situ* DRIFTS under both dark and illumination conditions."

WERE CHANGED TO:

"Despite these studies involving the heterogeneous oxidation of $SO_2$ under various conditions, the effects of co-existing pollutants on the heterogeneous oxidation of $SO_2$ under both dark and illuminated conditions need further investigation. Meanwhile, the interactions between organic and inorganic species in these heterogeneous processes at low concentrations are not fully understood. In this study, we focus on the effects of co-existing $NO_2$ and propene ($C_3H_6$) on the heterogeneous oxidation of $SO_2$ on $TiO_2$ under both dark and illuminated conditions with *in situ* DRIFTS."

Page 3, Line 28-31:

"Rather than UV lights, a xenon light is used for a better simulation of the UV irradiation from the sun on the earth's surface. Generally, our study could be helpful for gaining a better understanding of sulfate formation under complex air pollution conditions, in which abundant $SO_2$, $NO_x$, and VOCs as well as mineral dust exist in the atmosphere at the same time."

WERE CHANGED TO:

"Rather than UV lights, a xenon light is used in this study to better simulate the solar ultraviolet radiation on the earth's surface. Generally, our study could be helpful for gaining a better understanding of the heterogeneous formation of sulfate under complex air pollution conditions, in which abundant $SO_2$, $NO_x$, VOCs, and mineral dust coexist in the atmosphere."

*5. In the IC section of the methods, what column type was used? Moch et al., 2018 found that certain IC column types could easily separate hydroxymethanesulfonate (HMS) and sulfate and others could not. Since the author's method involves adding*

*a 1% formaldehyde solution to the samples, this would create HMS and possible an artifact in the IC measurements depending on the column type. Additionally, the authors mention that CH2O was observed when the surface was exposed to NO2 and C3H6, which might also indicate HMS formation*

*Moch, J. M., Dovrou, E., Mickley, L. J., Keutsch, F. N., Cheng, Y., Jacob, D. J., et al. (2018). Contribution of Hydroxymethane Sulfonate to Ambient Particulate Matter: A Potential Explanation for High Particulate Sulfur During Severe Winter Haze in Beijing. Geophysical Research Letters, 45(21), 11,969-11,979. https://doi.org/10.1029/2018GL079309*

**Response:** Thanks for the reminding. We used a Thermo AS14 Column in the IC, as we mentioned in the manuscript in the IC section. We read the paper of Moch and some references therein. But we had no clue if our column could separate HMS and sulfate or not, although in our IC measurements, we can see a peak of S(IV) a little earlier than S(VI). One good thing is that the IC measurements were only used to further compare the sulfate formation among different experimental systems in this study as the sulfate were also compared according to the *In situ* DRIFTS spectra and the K-M integrated area. The possible interferes of HMS on sulfate measurement in the IC doesn't change our conclusions. $CH_2O$ was observed when the surface was exposed to $NO_2$ and $C_3H_6$, but its reaction with sulfite might not be significant on the surface since the RH is <1%. In the revised manuscript, the possible interferes of HMS on sulfate were added. The description related to the IC measurement results were modified.

[Figure]

Page 8, Line 2:

DELETE "quantitatively"

Page 8, Line 1:

"These results confirmed the enhancing effect of $NO_2$ on the heterogeneous oxidation of $SO_2$ under dark conditions and the inhibiting effect of $NO_2$ and $C_3H_6$ on heterogeneous photooxidation of $SO_2$."

WERE CHANGED TO:

"Since formaldehyde was added to inhibit the oxidation of sulfite to sulfate in the solution, there is a possibility that HMS would be generated in the solution and be measured as sulfate (Moch et al., 2018). However, the possible interference by HMS in the measurement of sulfate by IC will not influence our conclusions on the effects of $NO_2$ and $C_3H_6$, since the K-M integrated area of sulfate in the *In situ* DRIFTS spectra were also compared."

Title of Figure. 4:

ADD: "Since formaldehyde was added to inhibit the oxidation of sulfite to sulfate in the solution, there is a possibility that HMS would be generated in the solution and be measured as sulfate."

*6. Many parts of the Results and Discussion section are better suited for placement in the methods section (e.g. the first and third sentence of section 3.1.1, large parts of the first paragraph of 3.3, etc.). The authors should consider moving sentences that describe how the experiments were conducted to the methods section and focus only on the results in the results section.*

**Response:** Thanks for the suggestions. Corresponding reorganization and revision were made in the revised manuscript. The description of experiments were moved to 2.2.1.

Page 4, Line 24- Page 5, Line 3:

"To investigate heterogeneous sulfate formation in complex atmospheric pollution, *in situ* DRIFTS was used to analyze the products on particle surfaces in the reactions under different conditions. Two series of *in situ* DRIFTS experiments were carried out in this study. For the heterogeneous reaction of $SO_2$ under different gas conditions, the $TiO_2$ sample was initially flushed with the synthetic air at a total flow rate of 100 mL min$^{-1}$ for 2 h. The temperature was 303 K and the relative humidity was less than 1% in all experiments. Then the background spectra were recorded when they showed little change with time. After that, gas reactants, such as 200 ppb $SO_2$, 200 ppb $NO_2$ and 200 ppb $C_3H_6$, were introduced to the gas flow and then passed through the reaction chamber for 12 h. These experiments were carried out under both dark and with UV-Vis irradiation conditions. The other series of experiments were step-by-step exposure experiments for further investigation of the effects of $NO_2$ and $C_3H_6$ on the heterogeneous oxidation of $SO_2$ with UV-Vis irradiation. The concentrations of reactants in the step-by-step exposure experiments were changed from 200 ppb to 200 ppm to strengthen the signals of the products. These step-by-step exposure experiments all included three steps, namely, first exposing the particles to $NO_2$, $C_3H_6$, or both for 2 h, then flushing with air for 1 h, and finally exposing them to $SO_2$ for 2 h."

*Other comments:*

*1. Line 12 on page 2 says that "SO2 can be irreversibly converted into sulfite, bisulfite or sulfate." This is incorrect for sulfite and bisulfite. Even if the particular conditions of the particle mean that sulfite or bisulfite are stable, if conditions change the*

*SO2-HSO3–SO32- equilibrium can shift and the authors should therefore avoid the use of the word "irreversibly" as applied to HSO3- and SO32- formation.*

**Response:** Thanks for the reminder. Yes, the word "irreversibly" is not correct.

> Page 2, Line 12:
>
> The word "irreversibly" was DELETED in the revised manuscript.

*2. Line 14 on page 2 says the authors say "low concentrations (200 ppb)." Was this a typo and the authors meant to write ppt? If not and the authors may mean low for a laboratory setting, but this type of phrasing could be confusing to non-laboratory scientists who may be interested in the author's work since atmospheric propene concentrations are rarely more than a couple of ppb. Later the authors say they used pollutants are "close to ambient concentration" (line 28 page 8), but 200 ppb NO2 and SO2 is much higher than ambient concentrations of these pollutants even during the extremely severe winter haze in Beijing. The authors should either include reference values for the concentrations of these gases in the laboratory compared to the atmosphere, or drop the use of "low concentrations" or "ambient" all together.*

**Response:** Yes, the so called "low concentrations (200 ppb)" only means low for laboratory *In situ* DRIFTS study. To avoid confusing, we dropped the use of "low concentrations" and "ambient" in the revised manuscript.

> Page 3, Line 14:
>
> DELETE "at low concentrations (200 ppb)"
>
> Page 6, Line 15-17:
>
> "which is consistent with previous studies (Ma et al., 2008;Liu et al., 2012), in spite of ambient concentration levels of $SO_2$ and $NO_2$ being used in this study."
>
> WERE CHANGED TO:
>
> "which is consistent with previous studies (Ma et al., 2008;Liu et al., 2012), in spite of much lower concentration levels of $SO_2$ and $NO_2$ being used in this study."
>
> Page 11, Line 28-29:
>
> "It was found that the presence of $NO_2$ could enhance the heterogeneous formation of sulfate with pollutants at close to ambient concentrations"
>
> WERE CHANGED TO:
>
> "Coexisting $NO_2$ could enhance the heterogeneous formation of sulfate with much lower concentrations (200 ppb) relative to previous studies (~100 ppm)  (Ma et al., 2008;Liu et al., 2012;Zhao et al., 2018)."

*3. Line 18 on page 2 regarding states "NO2 was proposed to act as a catalyst to activate O2 in the oxidation." This was a bit confusing, but I assume this means that the authors mean NO2 catalyzed the oxidation of SO2 by O2. If that is correct the*

*authors should change the sentence. Since there is also the heterogeneous oxidation of SO2 by NO2, the author be sure to clarify when the mechanisms involving NO2 they are referring to have SO2 oxidized by O2 and catalyzed by NO2 or have SO2 oxidized by NO2. I believe in most instances the authors are referring to the former reaction (i.e. catalyzed by NO2 and oxidized by O2).*

**Response:** Yes, we mean that $SO_2$ oxidized by $O_2$ and catalyzed by $NO_2$. To avoid confusing, we modified these sentences in the revised manuscript.

      Page 2, Line 18:

      "Therefore, $NO_2$ was proposed to act as an catalyst to activate $O_2$ in the oxidation"

      WERE CHANGED TO:

      "Therefore, $NO_2$ was proposed to act as a catalyst in the oxidation of $SO_2$ by $O_2$"

*4. With regards to the formation of hydroxymethanesulfonate (line 26-27 page 2), it would be appropriate for authors to also cite Moch et al., 2018 (referenced above) which also proposed the reaction of CH2O and sulfite/bisulfite in northern China winter haze.*

**Response:** This reference was ADDED accordingly.

      Page 2, Line 27:

      "HCHO was proposed to react with $SO_3^{2-}$ and generate hydroxymethanesulfonate (HMS) in the northern China winter haze period (Moch et al., 2018;Song et al., 2019)."

**Response for Reviewer #2**

*Chu et al. reported the effect of NO2 and C3H6 on the heterogeneous oxidation of SO2 into sulfate on TiO2 particles. Under dark conditions, the presence of NO2 generally enhanced the SO2 oxidation, whereas C3H6 had little influence. In contrast, the presence of NO2 and/or C3H6 suppressed the sulfate formation in the presence of UV irradiation. The authors attributed these results to the competitions between NO2 and SO2 for surface reactive sites on TiO2 and reactive oxygen species, and between C2H6 and SO2 for reactive oxygen species. However, their arguments on the underlying mechanisms are not satisfactorily explained based on the experimental results and the mechanistic insight is lacking. The impact of this study would be incremental to the understanding of heterogeneous oxidation of SO2 in the atmosphere. The manuscript requires major revisions before publication in ACP.*

**Response:** Thanks for the reviewer's comments. As mentioned in the response to the first reviewer, a discussion section about the proposed mechanisms on these effects are added in the revised manuscript.

Page 9- Page 11:

[revised manuscript text omitted]

*Specific comments:*

*The authors need to specify why TiO2 was chosen as the target material to put this work in a more appropriate context. For instance, the line 32 on page mention "TiO2, ...., has been widely used for studying heterogeneous photochemical reactions. What is the novelty in the present study?*

**Response:** Semiconductive metal oxides were known to be responsible for heterogeneous photo-oxidation of tracer gases. In this study, with the coexisting of multiple gases, the oxidation system is relatively complex, and both dark experiments and UV-Vis experiments were carried out. In order to better study the effects of $NO_2$ and $C_3H_6$ on the heterogeneous oxidation in the presence or absence of UV-Vis irradiation, we chose $TiO_2$ due to the fact that it is a semiconductor material and a well-known photocatalyst. The present study investigate the heterogeneous oxidation of $SO_2$ on $TiO_2$, but with quite different conditions in the gas phase in both dark and UV-Vis irradiation conditions, i.e. the coexisting of multiple gases including one VOC. Besides, compare the previous studies, relative low concentrations (200 ppb) are used for laboratory *In situ* DRIFTS study. In the revised manuscript, the following statement were added or modified:

Page 3, Line 13-19:

[revised manuscript text omitted]

*In the line 2 on page 5, I don't understand the correlation between the negative peaks (consumption) of the surface hydroxyls and the SO2 adsorption. Can you elaborate how the SO2 adsorption causes the negative peaks?*

**Response:** The two negative peaks at 3691 and 3630 cm$^{-1}$ were attributed to the vibration of hydroxyl on Ti atoms (Primet et al., 1971;Nanayakkara et al., 2012). The loss of surface hydroxyl groups from the surface upon adsorption of $SO_2$ implies that surface OH groups were involved in the reaction of $SO_2$ on $TiO_2$ in both dark reactions and UV-Vis irradiation conditions. This result is consistent with previous studies on $TiO_2$ (Nanayakkara et al., 2012;Ma et al., 2019). We made the following revision in the revised manuscript.

Page 5, Line 19-22:

"These negative peaks indicated that some $SO_2$ was absorbed on the surface hydroxyls, and were observed in all the reaction systems in this study, as shown in Fig. 1."

WERE CHANGED TO:

"These negative peaks were observed in all the reaction systems in this study, as shown in Fig. 1, which is consistent with previous studies (Nanayakkara et al., 2012;Ma et al., 2019). The loss of surface hydroxyl groups from the surface upon adsorption of $SO_2$ implies that surface OH groups were involved in the reaction of $SO_2$ on $TiO_2$ under both dark and UV-Vis irradiation conditions."

Page 9, Line 21-24:

ADD: "The heterogeneous oxidation of $SO_2$ on $TiO_2$ has been investigated by many previous studies. The following mechanisms for $SO_2$ adsorption on $TiO_2$ surfaces have been proposed in previous studies (Nanayakkara et al., 2012):

$$Ti-OH + SO_2 \rightarrow Ti-OSO_2H \tag{1}$$

$$2Ti-OH + SO_2 \rightarrow Ti_2-SO_3 \cdot H_2O \tag{2}$$"

*For the comparison of experiments with SO2 alone in the presence and the absence of UV illumination, the authors suggested the potential formation of molecularly adsorbed water, but the connection of this formation to the sulfate production is lacking. What is the role of molecularly adsorbed water in the reactions studied? The formation of adsorbed water is closely related to relative humidity (e.g., Romakkaniemi et al., 2001), but no information on relative humidity has been articulated in this work.*

*Romakkaniemi, S., Hämeri, K., Väkevä, M., and Laaksonen, A., J. Phys. Chem. A, 105, 8183−8188, 2001.*

**Response:** There are two sources of the molecularly adsorbed water related to sulfate formation. One is water formation during the heterogeneous reaction. For example, $SO_2$ reacted with two neighboring OH groups will result in sulfite and water formation, as shown in the following scheme (Nanayakkara et al., 2012). Meanwhile, a sulfate formation mechanism that involves both surface OH and O can also generate water, $HSO_3^- + O^- + OH^- \rightarrow SO_4^{2-} + H_2O + e^-$, as proposed by Zhang et al. (Zhang et al., 2006).

**Scheme 1** Reaction of $SO_2$ and surface hydroxyl groups to form adsorbed sulfite and water.

The other possible source of water is that water absorption from the introduced gas to the generated sulfate (Ma et al., 2019). In our experiments, no extra water flow was introduced to the reaction system (RH<1%), but still water cannot be entirely avoided in the introduced gas flows. In Fig.1, we can see the signal strength of the adsorbed water have good positive correlation with the amount of sulfite/sulfate formation in different experimental systems. In the revised manuscript, additional experimental information was added, and the discussion about the source of the adsorption water was extended.

Page 4, Line 27-28:

ADD: "The temperature was 303 K and the relative humidity was less than 1% in all experiments."

Page 5, Line 31 -Page 6, Line 4:

"Surface water may be formed in the photochemical reaction or via enhanced adsorption of water due to the increased hygroscopicity induced by sulfate (Ma et al., 2019)."

WERE CHANGED TO:

"Surface water can be formed in the heterogeneous reaction of $SO_2$ (Nanayakkara et al., 2012;Zhang et al., 2006), or via enhanced adsorption of water due to the increased hygroscopicity induced by sulfate (Ma et al., 2019). Although the RH was controlled at less than 1% in our experiments, water cannot be entirely removed in the introduced gas flows. In Fig.1, there is a positive correlation between the signal intensities of the adsorbed water and sulfite/sulfate among different experimental systems."

Page 9, Line 21-24:

ADD: "The heterogeneous oxidation of $SO_2$ on $TiO_2$ has been investigated by many previous studies. The following mechanisms for $SO_2$ adsorption on $TiO_2$ surfaces have been proposed in previous studies (Nanayakkara et al., 2012):

$$Ti - OH + SO_2 \rightarrow Ti - OSO_2H \qquad (1)$$
$$2Ti - OH + SO_2 \rightarrow Ti_2 - SO_3 \cdot H_2O \qquad (2)"$$

*The UV illumination (Fig. 1b) significantly enhanced the sulfate formation relative to the dark experiment. The authors need to discuss what is the role of UV illumination in the enhanced sulfate formation in the paragraph starting from the line 4 on page 5. Because of the lacking discussion, the argument in the lines 11-12 on page 5 does not seem correct. High sulfate formation rates under UV illumination might also cause the disappearance of sulfite peaks in the spectra due to rapid conversion of SO2 into sulfate.*

**Response:** We agree that UV illumination may also cause high sulfate formation rates and the disappearance of sulfite peaks in the spectra due to rapid conversion of $SO_2$ into sulfate. With UV illumination, $TiO_2$ can be excited by UV light ($\lambda < 387$ nm), resulting in additional ROS (primarily $O_2^-$ and $OH$), and oxidize more $SO_2$ to sulfate on $TiO_2$ than that under dark condition (Shang et al., 2010;Chen et al., 2012). We added some discussion on Page 5 as well as in the new discussion section.

Page 5, Line 26-29:

ADD: "With UV-Vis illumination, $TiO_2$ can be excited by UV light ($\lambda < 387$ nm), then the photogenerated electrons and holes can react with $H_2O$ and $O_2$ to produce additional ROS (primarily $\cdot O_2^-$ and $\cdot OH$), and oxidize more $SO_2$ to sulfate on $TiO_2$ than that produced under dark conditions (Shang et al., 2010a;Chen et al., 2012)."

Page 5, Line 11-12 (Old version):

DELETE "Compared with the reaction under dark conditions, i.e. Fig.1 (a), sulfate species rather than sulfite species were generated, indicating a different mechanism for the formation of sulfate with UV irradiation."

Page 10, Line 18-28:

ADD: "With UV illumination, $TiO_2$ can be excited by UV light ($\lambda < 387$ nm), then the photogenerated electrons and holes can react with $H_2O$ and $O_2$ to produce additional ROS (primarily $\cdot O_2^-$ and $\cdot OH$), and oxidize more $SO_2$ to sulfate on $TiO_2$ than that produced under dark conditions (Shang et al., 2010a;Chen et al., 2012).The detailed mechanism was summarized by Chen et al. (Chen et al., 2012) and references therein:

$$TiO_2 + h\nu(\lambda < 387 \text{ nm}) \rightarrow e^-h^+ \rightarrow e^- + h^+ \qquad (9)$$

$$O_2 + e^- \rightarrow \cdot O_2^- \qquad (10)$$

$$H_2O + h^+ \rightarrow \cdot OH + H^+ \qquad (11)$$

Then the $SO_2$ can react with these ROS and promote the formation of sulfate (Shang et al., 2010b):

$$Ti - SO_2 + \cdot O_2^- \rightarrow Ti - SO_3 + O^- \qquad (12)$$

$$Ti - SO_3 + H_2O \rightarrow Ti - H_2SO_4 \qquad (13)$$

$$Ti - SO_3^{2-} + 2 \cdot OH \rightarrow Ti - SO_4^{2-} + H_2O \qquad (14)"$$

*In the lines 28-30 on page 5, the authors described that more sulfate with UV irradiation in the SO2 + NO2 system than without UV irradiation was consistent with the results in the SO2 alone system. However, the mechanism might be different between the two systems. For instance, nitrate formed is subjected to photolysis under UV irradiation (> 300 nm). Recent work has found that nitrate photolysis can enhance the conversion of SO2 into sulfate in wet aerosols (Gen et al., 2019). It would be useful if the authors provide more discussion from a perspective of the mechanisms.*

*Gen, M., Zhang, R., Huang, D., Li, Y., and Chan, C. K., Heterogeneous Oxidation of SO2 in Sulfate Production During Nitrate Photolysis at 300 nm: Effect of pH, Relative Humidity, Irradiation Intensity, and the Presence of Organic Compounds. Environ. Sci. Technol., 2019.*

*In the line 31 on page 5, it is not clear about what is the opposing effect. The authors need to clarify this effect.*

**Response:** Thanks for the reminding. Since nitrate were generated in the presence of $NO_2$, there is a possibility that photolysis of the nitrate would enhance sulfate formation. The RH is very low (RH<1%) in our experiments. It seems the enhancing effect of nitrate on sulfate formation is not very significant since sulfate formation in $SO_2$+$NO_2$ system is less than that of $SO_2$ alone system with the presence of UV irradiation. We added the discussion about this effect and modified the related description in the revised manuscript.

Page 6, Line 20-23:

"Compared to the dark experiment of $SO_2$ and $NO_2$ in Fig 1(c), more sulfate species were generated with UV irradiation, which is consistent with the fact that UV irradiation significantly promotes sulfate formation in the reaction of $SO_2$ alone. Also, compared with the spectra of $TiO_2$ exposed to only $SO_2$ with UV irradiation, the bands of sulfate species decreased in intensity in the presence of $NO_2$. The effect of $NO_2$ on sulfate formation with UV irradiation was opposite to that under dark conditions."

WERE CHANGED TO:

"Compared to the dark experiment of $SO_2$ and $NO_2$ in Fig 1(c), more sulfate species were generated with UV-Vis irradiation, which might be due to the fact that UV-Vis irradiation significantly promotes sulfate formation by generating additional active species (Shang et al., 2010a;Chen et al., 2012) as in the reaction of $SO_2$ alone."

Page 10, Line 21-Page 11, Line 2:

ADD: "In the UV-Vis irradiation experiments, $NO_2$ had a distinct suppressing effect on the sulfate formation compared to the individual reaction of $SO_2$. Rather than resulting in ROS formation and oxidation of S(IV) to S (VI) in dark experiments, the main reaction of $NO_2$ with the surface ROS resulted in nitrate formation in experiments with UV-Vis irradiation (Ndour et al., 2008;Yu and Jang, 2018).

$$Ti - NO_2 + \cdot OH \rightarrow Ti - HONO_2 \qquad (15)$$
$$Ti - NO_2 + \cdot O_2^- \rightarrow Ti - NO_2^- + O_2 \qquad (16)$$

The nitrate or nitrite generated from the oxidation of $NO_2$ might block some surface reactive sites, since in the step-to-step experiments, the pre-adsorption of $NO_2$ on $TiO_2$ also suppressed the formation of sulfate and resulted in similar sulfate formation to that in the experiment introducing $NO_2$ and $SO_2$ simultaneously. The competition between $SO_2$ and $NO_2$ for surface reactive sites might be the main reason for the fact that the coexistence of $NO_2$ with $SO_2$ resulted in decreased sulfate formation with UV-Vis irradiation in this study. Although Gen et al. (Gen et al., 2019) found that photolysis of nitrate enhanced sulfate formation in wet aerosols, this mechanism may not be applied in this study since the reaction system is quite different from their study. The ROS which oxidize S(IV) to S(VI) are mainly $\cdot O_2^-$ and $\cdot OH$ in the presence of UV-Vis irradiation rather than the photolysis of nitrate."

*The discussion in the lines 3-5 on page 7 cannot explain the suppressing effect of NO2 on the sulfate formation under UV irradiation, relative to the SO2 alone system. How the presence of NO2 suppresses sulfate formation on TiO2 under UV irradiation.*

**Response:** Thanks for your reminding. We added the possible blocking of surface reactive sites by nitrate here.

Page 7, Line 31-Page 8, Line 1:

Add "What's more, the nitrate formation from oxidation of $NO_2$ might block some surface reactive sites, and therefore, resulted in less sulfate formation in the reaction of $SO_2+NO_2$ than that of $SO_2$ alone with UV-Vis irradiation."

*Reactive oxygen species (ROS) are seemingly responsible for the oxidation of SO2 into sulfate. However, the lacking information here is what ROS are generated on TiO2. The authors need to explain what are formed on TiO2 and how the formed ROS oxide SO2 in the earlier part of the manuscript.*

**Response:** More discussion about the formation and their roles in oxidation of $SO_2$ were added in the revised manuscript.

Page 5, Line 26-29:

"$TiO_2$ can be excited by UV light ($\lambda < 387$ nm), resulting in active species (primarily $O_2^-$ and OH) that can participate in atmospheric photochemical reactions (Chen et al., 2012)."

WERE CHANGED TO:

"With UV-Vis illumination, $TiO_2$ can be excited by UV light ($\lambda < 387$ nm), then the photogenerated electrons and holes can react with $H_2O$ and $O_2$ to produce additional ROS (primarily $\cdot O_2^-$ and $\cdot OH$), and oxidize more $SO_2$ to sulfate on $TiO_2$ than that produced under dark conditions (Shang et al., 2010a;Chen et al., 2012)."

Page 7, Line 19-24:

ADD: "The presence of $NO_2$ seemed to induce the generation of some ROS, which oxidize S(IV) to S(VI) on $TiO_2$ (Ma et al., 2008;Liu et al., 2012;Ma et al., 2017). The detailed mechanism for this effect has not been fully explored and will be discussed later. It has also been proposed that aqueous oxidation of $SO_2$ by $NO_2$ (as an oxidizing agent) contributed to significant sulfate formation in haze events (Wang et al., 2016b;Cheng et al., 2016). This reaction should not be the main pathway in the reaction systems in this study since the experiments were carried out under dry conditions (RH<1%), although water can still exist, as we mentioned earlier."

Page 9, Line 26-Page 10, Line 6:

ADD: "It has been demonstrated that coexisting $NO_2$ can induce the generation of some ROS, which oxidize S(IV) to S(VI) on mineral oxides (Ma et al., 2008;Liu et al., 2012;Ma et al., 2017). There were several possible responsible ROS proposed in previous studies, although the detailed mechanism has not yet been fully explored. One possible ROS is $N_2O_4$, which can undergo hydrolysis to N(III) and N(V) species (Liu et al., 2012;Finlayson-Pitts et al., 2003;Li et al., 2018). These reactive nitrogen species can oxidize S(IV) to S(VI) (Wang et al., 2016b;Li et al., 2018).

$$2Ti - NO_2 \rightarrow Ti_2 - N_2O_4 \qquad\qquad (4)$$

$$N_2O_4(ad) \rightarrow NO^+NO_3^- \xrightarrow{H_2O} HNO_3 + HONO \qquad\qquad (5)$$

Besides $N_2O_4$, $NO_2$ may also react directly with surface OH and form $HNO_3$ on $TiO_2$ (Liu et al., 2017a). The $HNO_3$ generated through this pathway may also contribute to the oxidation of S(IV) to S(VI). It has also been proposed that aqueous oxidation of $SO_2$ by $NO_2$ (as an oxidizing agent) contributed to significant sulfate formation in haze events (Wang et al., 2016b;Cheng et al., 2016). This aqueous reaction should not be significant in the reaction systems of this study due to the limited amount of water under low RH condition (<1% RH)."

*In the lines 19−21 on page 8, this discussion is purely a simple guess. What are potential products blocking the surface reactive sites? Do you have any experimental evidence to support the presence of products (e.g., in DRIFTS spectra)?*

**Response:** In the step-to-step experiments, since the $NO_2$ and $C_3H_6$ was cut off after pre-adsorption, and ROS was expected to be generated on $TiO_2$ with UV, it is quite possible that some products blocked surface reactive sites and decreased sulfate formation. The potential products would be nitrate, aldehydes and carboxylic acids from the oxidation of $NO_2$ and propene. Besides the abundant nitrate, the bands at 1750 and 1524 cm$^{-1}$ in DRIFTS spectra, which could be assigned to $CH_2O$ (Liao et al., 2001) and COO groups (Mattsson and Österlund, 2010), respectively. We add some additional discussion in the revised manuscript.

Page 9, Line 14-18:

"The detailed reason for this phenomenon was not discovered in this study. One possible reason might be that some products were generated when the particles were exposed to $NO_2$ and $C_3H_6$ at the same time, and these species seemed to block some reactive sites on $TiO_2$ and suppress sulfate formation in heterogeneous photooxidation."

WERE CHANGED TO:

"Although the detailed reason for this phenomenon was not discovered in this study, a possible reason might be that the oxidation products from $NO_2$ and $C_3H_6$ blocked some reactive sites on $TiO_2$ and suppress sulfate formation in heterogeneous photooxidation, since $NO_2$ and $C_3H_6$ was cut off after pre-adsorption and ROS was expected to be generated on $TiO_2$ with UV-Vis irradiation. According to the DRIFTS spectra in Fig. 4(c), besides nitrate, aldehydes (1750 cm$^{-1}$) and carboxylic acids (1524 cm$^{-1}$) were also observed on $TiO_2$ after the pre-adsorption with of $NO_2 + C_3H_6$."

*In the line 26 on page 8, what was saturated with?*

**Response:** It is sulfite.

Page 11, Line 26-27:

"With reaction time increasing, the surface became saturated and prevented SO$_2$ from adsorbing on the particles further"

WERE CHANGED TO:

"With reaction time increasing, the adsorption sites on the surface became saturated with sulfite and prevented SO$_2$ from adsorbing on the particles further."

*In the line 11 on page 9, how can we know that NO2 may compete with SO2 for both surface active sites and ROS? Based on the results (Fig. 6), NO2 appears to compete with SO2 for the surface active sites, but not for ROS since the sulfate formation rate (increasing rate of the K-M integrated area) with the step-by-step gas (NO2 first and then SO2) injection in the later reaction time becomes comparable to that with both gases together.*

**Response:** We agree with the reviewer that NO$_2$ mainly compete with SO$_2$ for the surface active sites according to the experiments result. In the step-by-step (NO$_2$ first and then SO$_2$) reaction, sulfate formation was less than that in the reaction of NO$_2$+SO$_2$, while in the later reaction time becomes comparable to that in the reaction of NO$_2$+SO$_2$, indicating that NO$_2$ maily compete with SO$_2$ for the surface active sites and resulted in less sulfate formation compare to the reaction of SO$_2$ alone with the presence of UV-Vis irradiation which resulted in continuous production of ROS on TiO$_2$ surface. We deleted the related description in the Conclusion section, and discussed this in the discussion part in the revised manuscript.

Page 11, Line 27 (Page 9, Line 8-13 in the Old version):

DELETE: "In the step-by-step experiments, presaturation by C$_3$H$_6$ and then flushing had no significant influence on sulfate formation in the heterogeneous photooxidation of SO$_2$, while presaturation with NO$_2$ and then flushing suppressed sulfate formation. These results indicated that C$_3$H$_6$ mainly competes with SO$_2$ for ROS on the surface, while NO$_2$ competes with SO$_2$ for both surface active sites and ROS. The coexistence of NO$_2$ and C$_3$H$_6$ seemed to lead to more organics formation on the surface of TiO$_2$ and suppressed sulfate formation more compared to introducing only one of them."

Page 10, Line 29-Page 11, Line 7:

ADD: "In the UV-Vis irradiation experiments, NO$_2$ had a distinct suppressing effect on the sulfate formation compared to the individual reaction of SO$_2$. Rather than resulting in ROS formation and oxidation of S(IV) to S (VI) in dark experiments, the main reaction of NO$_2$ with the surface ROS resulted in nitrate formation in experiments with UV-Vis irradiation (Ndour et al., 2008;Yu and Jang, 2018).

$$Ti - NO_2 + \cdot OH \rightarrow Ti - HONO_2 \qquad (15)$$
$$Ti - NO_2 + \cdot O_2^- \rightarrow Ti - NO_2^- + O_2 \qquad (16)$$

The nitrate or nitrite generated from the oxidation of NO$_2$ might block some surface reactive sites, since in the step-to-step experiments, the pre-adsorption of NO$_2$ on TiO$_2$ also suppressed the formation of sulfate and resulted in similar sulfate formation to that in the experiment introducing NO$_2$ and SO$_2$ simultaneously. The competition between SO$_2$ and NO$_2$ for

surface reactive sites might be the main reason for the fact that the coexistence of $NO_2$ with $SO_2$ resulted in decreased sulfate formation with UV-Vis irradiation in this study."

*The statement in the line 14 on page 9 is too general. Need to rewrite.*

**Response:** This statement was rewritten.

Page 12, Line 3-4:

"These results indicated that heterogeneous oxidation of $SO_2$ might be influenced by a number of factors under complex pollution conditions with various gas pollutants."

WERE CHANGED TO

"These results indicated that heterogeneous oxidation of $SO_2$ might be influenced by the co-existing inorganic and organic gas pollutants under complex pollution conditions due the competition for ROS and active surface sites among them."

Page 12, Line 5:

DELETE "Besides inorganic species, organics could also significantly change the heterogeneous oxidation of $SO_2$."

*Minor comments:*

*In the line 30 on page 2, please specify what type of illumination the authors refer to.*

**Response:** It is UV illumination.

Page 2, Line 29:

"Illumination"

WERE CHANGED TO

"UV illumination"

*Line 9 on page 4: what are the wavelengths of the UV irradiation?*

**Response:** The information about the wavelengths was added in the revised manuscript.

Page 4, Line 19-21:

ADD "The wavelengths of the UV-Vis irradiation were measured to be in the range of 250-850 nm by a fiber optic spectrometer (BLUE-Wave-UVNb, Stellar Net Inc., USA), as shown in Fig. S1 in the Supplemental Information."

*Please state gas concentrations in the experimental.*

**Response:** The information about the concentrations was added in the revised manuscript.

Page 4, Line 29-30:

ADD "After that, gas reactants, such as 200 ppb $SO_2$, 200 ppb $NO_2$ and 200 ppb $C_3H_6$, were introduced to the gas flow and then passed through the reaction chamber for 12 h."

Page 5, Line 14; Page 6, Line 8 & Line 18:

ADD "DRIFTS spectra for heterogeneous reaction of 200 ppb $SO_2$ on $TiO_2$"

ADD" 200 ppb $SO_2$ and 200 ppb $NO_2$"

ADD" 200 ppb $SO_2$ and 200 ppb $NO_2$"

*I believe that the title of sub-section 3.1.3 is typo.*

**Response:** Thanks for pointing out. "$NO_2$" was REVISED to "$C_3H_6$".

*Line 17 on page 8: "with of" is typo.*

**Response:** Thanks for pointing out. The word "of" was DELETED.

**Response for Reviewer #3**

*This manuscript presents an experimental study on the influence of NO2 and a specific VOC (propene) on the heterogeneous production of sulfate on TiO2 particles. The study argues for the complexity in the situation of multiple precursors. The topic fits well in the journal. However, there are significant issues within the manuscript. Below are the major, minor and technical comments. They should be satisfactorily addressed before consideration for publication in the final ACP.*

*Major:*

*A major question that I have is on the set up of the experiments in which many details are missing in the current manuscript. Specifically, (1) is relative humidity controlled? A lot of previous studies show the importance of RH in heterogeneous reactions. RH (or the abundance of water vapor) also impacts gas phase reactions through HOx cycle. (2) about UV light illumination. What is the amplitude and the range of wavelength? Is it represent of the real atmosphere? (3) the detection of ion chromatography. Is it interfered by HMS hydroxymethanesulfonate? (4) Rational of the choice of materials: TiO2 and propene. How well do they represent the aerosol particles and VOCs? These above questions should be clearly answered in the manuscript.*

**Response:** Thanks for the reviewer's comments. More details were added in the revised manuscript as the reviewer suggested.

(1) RH.

In our experiments, no water flow was introduced to the reaction system (RH<1%). In the revised manuscript, additional experimental information was added, and the discussion about the source of the adsorption water was extended.

Page 4, Line 27-28:

ADD: "The temperature was 303 K and the relative humidity was less than 1% in all experiments."

(2) UV.

In the manuscript, we mentioned that "The intensity of UV irradiation was measured as 478 $\mu W\ cm^{-2}$", while the information about the wavelengths was added in the revised manuscript. We also compared the spectrum of the UV irradiation with solar irradiation on the earth surface, as shown in the below picture, which was also added in the Supplemental Information. The spectrum of the UV-Vis irradiation seems to be comparable with the spectrum of solar irradiation on the earth surface, and therefore we think the UV-Vis irradiation used in this study may represent the real atmosphere.

Page 4, Line 19-21:

ADD "The wavelengths of the UV-Vis irradiation were measured to be in the range of 250-850 nm by a fiber optic spectrometer (BLUE-Wave-UVNb, Stellar Net Inc., USA), as shown in Fig. S1 in the Supplemental Information. The spectrum of the UV-Vis irradiation is comparable to the spectrum of solar irradiation on the earth surface, and therefore we think the UV-Vis irradiation used in this study may represent the conditions in the real atmosphere."

[Figure]

(3) IC.

Thanks for the reminding. We used a Thermo AS14 Column in the IC, which was mentioned in the manuscript in the IC section. We read the paper of Moch and some references therein. But we had no clue if our column could separate HMS and sulfate or not, although in our IC measurements, we can see a peak of S(IV) a little earlier than S(VI). One good thing is that

the IC measurements were only used to further compare the sulfate formation among different experimental systems in this study as the sulfate were also compared according to the *In situ* DRIFTS spectra and the K-M integrated area. The possible interferes of HMS on sulfate measurement in the IC doesn't change our conclusions. $CH_2O$ was observed when the surface was exposed to $NO_2$ and $C_3H_6$, but its reaction with sulfite might not be significant on the surface since the RH is <1%. In the revised manuscript, the possible interferes of HMS on sulfate were added. The description related to the IC measurement results were modified.

[Figure]

Page 8, Line 2:

DELETE "quantitatively"

Page 8, Line 1:

[revised manuscript text omitted]

*The second one is on the structure of the manuscript. Currently, a big chunk of the method description resides in the results and discussion. I suggest that the authors should re-organize the structure and separate method, results, and discussion (three sections). The experiments conducted in this study should be summarized at first in the method section. In the discussion section, a more thorough and clear discussion on the influencing factors of SO2 oxidation should be provided.*

**Response:** Thanks for the suggestions. Corresponding reorganization and revision were made in the revised manuscript. The description of experiments were moved to 2.2.1. A separate discussion part was also added after the results section.

Page 4, Line 24- Page 5, Line 3:

[revised manuscript text omitted]

*The third one is on the proposed mechanisms which in my opinion are not well justified. The study intends to explore the underlying mechanisms through different combinations of chemical precursors. The proposed mechanisms are specifically related to the production and/or competition for ROS and surface reactive sites. But the study does not provide a good way in*

*the experiments to argue for the importance of ROS and reactive sites. What are differences in terms of production and fate of ROS under dark and illumination conditions? Is there a way to detecting the saturation of surface reactive sites?*

**Response:** As mentioned in the response of the second major comment, we added a separate discussion section to discuss the proposed mechanisms of sulfate formation and the competition for ROS and surface reactive sites. Some of the related discussion are listed below. Although we carried our step-to-step experiments and trying to better understand the competition for ROS and surface reactive sites, there is still uncertainty about these process. Take $NO_2$ as an example, in the step-by-step ($NO_2$ first and then $SO_2$) reaction, sulfate formation was less than that in the reaction of $NO_2+SO_2$, while in the later reaction time becomes comparable to that in the reaction of $NO_2+SO_2$, indicating that $NO_2$ maily compete with $SO_2$ for the surface active sites and resulted in less sulfate formation compare to the reaction of $SO_2$ alone with the presence of UV-Vis irradiation which resulted in continuous production of ROS on $TiO_2$ surface. We deleted the related description in the conclusion section, and discussed this in the discussion part in the revised manuscript. As the reviewer pointed out, detecting the saturation of surface reactive sites, as well as the ROS should be greatly helpful, but unfortunately, we don't have a good method for these measurement yet in our group.

Page 12, Line 2 (Page 9, Line 8-13 in the Old version):

[revised manuscript text omitted]

*Minor:*

*Page 5, Line 9-10: Elaborate on the processes leading to surface water formation.*

**Response:** There are two sources of the molecularly adsorbed water related to sulfate formation. One is water formation during the heterogeneous reaction. For example, $SO_2$ reacted with two neighboring OH groups will resulted in sulfite and water formation, as shown in the following scheme (Nanayakkara et al., 2012). Meanwhile, a sulfate formation mechanism that involves both surface OH and O can also generate water, $HSO_3^- + O^- + OH^- \rightarrow SO_4^{2-} + H_2O + e^-$, as proposed by Zhang et al. (Zhang et al., 2006).

[Figure]

**Scheme 1**    Reaction of $SO_2$ and surface hydroxyl groups to form adsorbed sulfite and water.

The other possible source of water is that water absorption from the introduced gas to the generated sulfate (Ma et al., 2019). In our experiments, no water flow was introduced to the reaction system (RH<1%), but still water cannot be entirely avoided in the introduced gas flows. In Fig.1, we can see the signal strength of the adsorbed water have good positive correlation with the amount of sulfite/sulfate formation in different experimental systems. In the revised manuscript, additional experimental information was added, and the discussion about the source of the adsorption water was extended.

Page 9, Line 21-24:

ADD: "The heterogeneous oxidation of $SO_2$ on $TiO_2$ has been investigated by many previous studies. The following mechanisms for $SO_2$ adsorption on $TiO_2$ surfaces have been proposed in previous studies (Nanayakkara et al., 2012):

$$Ti - OH + SO_2 \rightarrow Ti - OSO_2H \qquad\qquad\qquad (1)$$
$$2Ti - OH + SO_2 \rightarrow Ti_2 - SO_3 \cdot H_2O \qquad\qquad\qquad (2)"$$

Page 5, Line 31 -Page 6, Line 4:

"Surface water may be formed in the photochemical reaction or via enhanced adsorption of water due to the increased hygroscopicity induced by sulfate (Ma et al., 2019)."

WERE CHANGED TO:

"Surface water can be formed in the heterogeneous reaction of $SO_2$ (Nanayakkara et al., 2012;Zhang et al., 2006), or via enhanced adsorption of water due to the increased hygroscopicity induced by sulfate (Ma et al., 2019). Although the RH was controlled at less than 1% in our experiments, water cannot be entirely removed in the introduced gas flows. In Fig.1, there is a positive correlation between the signal intensities of the adsorbed water and sulfite/sulfate among different experimental systems."

*Page 9, Paragraph 2: Elaborate on the different effects of different VOCs from previous studies.*

**Response:** More discussion about the effects of different VOCs on sulfate formation was added to the paragraph.

Page 12, Line 3-12:

"These results indicated that heterogeneous oxidation of $SO_2$ might be influenced by a number of factors under complex pollution conditions with various gas pollutants. Besides inorganic species, organics could also significantly change the heterogeneous oxidation of $SO_2$. In this study, only one VOC was investigated, while the heterogeneous oxidation of various VOCs has been reported in previous studies (Niu et al., 2017;Du et al., 2000). The competition for ROS and surface reactive sites between these VOCs and $SO_2$ is likely to suppress sulfate formation in the heterogeneous reactions. Due to the different properties of the oxidation products, the influence of coexisting VOCs might be different for different VOC species."

WERE CHANGED TO:

"These results indicated that heterogeneous oxidation of $SO_2$ might be influenced by the co-existing inorganic and organic gas pollutants under complex pollution conditions due the competition for ROS and active surface sites among them. In this study, only one VOC was investigated, while the heterogeneous oxidation of various VOCs has been reported in previous studies (Niu et al., 2017;Du et al., 2000). When a VOC and $SO_2$ coexist, the competition for ROS and surface reactive sites between the VOC and $SO_2$ is likely to suppress sulfate formation in the heterogeneous reactions, such as that observed for the presence of $CH_3CHO$ on $\alpha$-$Fe_2O_3$ in dark experiments (Zhao et al., 2015), the presence of $C_7H_{16}$ on $TiO_2$ with UV-Vis irradiation (Du et al., 2000), and the presence of $C_3H_6$ on $TiO_2$ under dark condition or with UV-Vis irradiation in this study. Due to the different properties of the oxidation products, the influence of coexisting VOCs might be different for different VOC species and on different mineral dusts. Some coexisting VOCs, such as HCOOH on $\alpha$-$Fe_2O_3$ (Wu et al., 2013), and HCHO in aerosol water (Moch et al., 2018;Song et al., 2019) might enhance sulfate formation."

*The authors may consider move Figure 2 to the supplemental.*

**Response:** Fig.2 was REMOVED to the Supplemental Information.

*Technical:*

*Page 1, Line 17: full expression for "DRIFTS"*

**Response:** The full expression for FRIFTS (Diffuse Reflectance Infrared Fourier Transform Spectroscopy) was ADDED in the abstract. There was also a full expression in the third paragraph in the introduction.

*Page 2, Line 5: "the mechanisms of heterogeneous reaction processes as well as their"*

**Response:** Thanks! It was REVISED accordingly.

[revised manuscript text omitted]
. TheseOHcanthenreactions,izeand result in much moreformationexperiments withUV, although individual $C_3H_6$ has little effect on sulfate formationdue to the competitionforor the available ROS. In the step by step experiments, presaturation by $C_3H_6$ and then flushing had no significant influence on sulfate formation in the heterogeneous photooxidation of $SO_2$, while presaturation with $NO_2$ and then flushing suppressed sulfate formation. However, after about 2 hours of reaction, sulfate formation on $TiO_2$ pre-saturated with $NO_2$ became comparable with the experiment with $SO_2$ and $NO_2$ together. 
[revised manuscript text omitted]